# A proteomic landscape of diffuse-type gastric cancer

Sai Ge [1,2], Xia Xia[1], Chen Ding[1,3], Bei Zhen[1], Quan Zhou[1], Jinwen Feng[1,4], Jiajia Yuan[1,2], Rui Chen[5], Yumei Li[5], Zhongqi Ge[5], Jiafu Ji[2], Lianhai Zhang [2], Jiayuan Wang[2], Zhongwu Li[2], Yumei Lai[2], Ying Hu[2], Yanyan Li[2], Yilin Li[2], Jing Gao[2], Lin Chen[6], Jianming Xu[7], Chunchao Zhang[8], Sung Yun Jung[8], Jong Min Choi[8], Antrix Jain[8], Mingwei Liu[1], Lei Song[1], Wanlin Liu[1], Gaigai Guo[1], Tongqing Gong[1], Yin Huang[1], Yang Qiu[1], Wenwen Huang[1,2], Tieliu Shi[4], Weimin Zhu[1], Yi Wang[1,8], Fuchu He[1,3], Lin Shen[1,2] & Jun Qin[1,3,8]

The diffuse-type gastric cancer (DGC) is a subtype of gastric cancer with the worst prognosis and few treatment options. Here we present a dataset from 84 DGC patients, composed of a proteome of 11,340 gene products and mutation information of 274 cancer driver genes covering paired tumor and nearby tissue. DGC can be classified into three subtypes (PX1–3) based on the altered proteome alone. PX1 and PX2 exhibit dysregulation in the cell cycle and PX2 features an additional EMT process; PX3 is enriched in immune response proteins, has the worst survival, and is insensitive to chemotherapy. Data analysis revealed four major vulnerabilities in DGC that may be targeted for treatment, and allowed the nomination of potential immunotherapy targets for DGC patients, particularly for those in PX3. This dataset provides a rich resource for information and knowledge mining toward altered signaling pathways in DGC and demonstrates the benefit of proteomic analysis in cancer molecular subtyping.

[1] The Joint Laboratory of Translational Medicine, National Center for Protein Sciences (Beijing) and Peking University Cancer Hospital, State Key Laboratory of Proteomics, Institute of Lifeomics, Beijing 102206, China. [2] Key Laboratory of Carcinogenesis and Translational Research (Ministry of Education), Peking University Cancer Hospital & Institute, Beijing 100142, China. [3] State Key Laboratory of Genetic Engineering, Human Phenome Institute, Institutes of Biomedical Sciences, and School of Life Sciences, Zhongshan Hospital, Fudan University, Shanghai 200433, China. [4] Center for Bioinformatics, East China Normal University, Shanghai, 200241 China. [5] Human Genome Sequencing Center, Department of Molecular and Human Genetics, Baylor College of Medicine, Houston, TX 77030, USA. [6] General Hospital of Chinese People's Liberation Army, Beijing 100853, China. [7] Affiliated Hospital of Academy of Military Medical Sciences, Beijing 100071, China. [8] Alkek Center for Molecular Discovery, Verna and Marrs McLean Department of Biochemistry and Molecular Biology, Department of Molecular and Cellular Biology, Baylor College of Medicine, Houston, TX 77030, USA. These authors contributed equally: Sai Ge, Xia Xia, Chen Ding, Bei Zhen. Correspondence and requests for materials should be addressed to F.H. (email: hefc@nic.bmi.ac.cn) or to L.S. (email: linshenpku@163.com) or to J.Q. (email: jqin1965@126.com)

Gastric cancer (GC) is the third leading cause of cancer mortality in the world, particularly in East Asia, which accounts for more than half of the cases worldwide[1, 2]. GC is heterogeneous at the genetic and cellular levels[3]. The Lauren classification stratifies GC into diffuse, intestinal, and mixed type and such classification is widely used in the clinics[4, 5]. The diffuse-type gastric cancer (DGC) accounts for approximately 30% of GC, and has poor clinical outcome with few targeted treatment options[6, 7]. Previous study by The Cancer Genome Atlas (TCGA) project mapped a genomic landscape of GC and classified GC into four subtypes, namely Epstein−Barr virus positive (EBV), microsatellite instable (MSI), genome stable (GS), and chromosomal instability (CIN)[8]. DGC was classified mainly as genomically stable tumors. A study from the Asian Cancer Research Group (ACRG) clarified GC based on gene expression data into four subtypes associated with distinct clinical outcomes: MSI, MSS/EMT (microsatellite stable and epithelial-to-mesenchymal transition), MSS/TP53+ (TP53 active), and MSS/TP53− (TP53 inactive). The MSS/EMT subtype containing mainly DGC showed the worst prognosis[9]. Similar studies identified GC driver genes and dysregulated pathways at genomic and transcriptional levels, which have significantly enhanced our understanding of gastric cancer[10–13].

Realizing proteins as the "executioners of life" that determine phenotype, the Clinical Proteomic Tumor Analysis Consortium (CPTAC) has published integrated analyses, including DNA methylation, copy number alterations (CNVs), and mRNA and protein profiling of TCGA tumor specimens to portray proteogenomic landscapes of colorectal cancer, breast, and ovarian cancers[14–16]. The CPTAC work found that, in general, messenger RNA (mRNA) transcript abundance does not reliably predict protein abundance differences, and although CNV showed strong cis- and trans-effects on mRNA abundance, relatively few of these differences extend to the protein level[14, 16]. As a result, subtyping of tumors from nuclear acid-based genomic data did not fully agree with subtyping from proteomic data. Nevertheless, integrated proteogenomic analyses afforded a new paradigm for understanding cancer biology with functional context to interpret genomic data.

Ideally, paired tumor and nearby/normal tissue from the same patient should be compared to find genetic and proteomic alterations. So far, published CPTAC studies analyzed tumor samples from many patients, but profiled few normal tissues from separate test subjects as normal controls. While this is not an issue for genomic analysis, the heterogeneity introduced by variation from different test subjects can further complicate proteomics analyses and limit our ability to profile individually altered cancer proteome, and identify dysregulated-signaling pathways that can be tailored for individualized medicine.

Here, we present the first proteomic analysis of DGC by measuring 84 pairs of tumors and their nearby tissues. Based on proteomic results alone, DGCs can be classified into three subtypes with distinct pathway enrichment and clinical outcomes. Our study provides a rich resource for data mining and guidance for clinical validation.

## Results

**Proteome profiling and targeted exome sequencing.** We examined 2451 GC cases deposited in the tumor tissue bank of Beijing Cancer Hospital and selected 84 diffuse-type tumors (T) together with their matching nearby tissues (N) that met our criteria for proteome profiling and targeted exome sequencing (Supplementary Fig. 1; Supplementary Data 1). We employed a fast mass spectrometry (MS) workflow for proteome profiling (fast-seq)[17]. Extracted proteins from frozen tissues were digested

with trypsin and the resulting peptides were separated at high pH with small columns packed in pipette tips (sRP); fractionated peptides were pooled to 6 MS runs using a 75 min high-performance liquid chromatography gradient at low pH. Such a workflow allowed the analysis of a proteome in half a day (Supplementary Fig. 1c). The MS platform was stable and repeatable as judged by quality control runs during the entire data-collecting period (Supplementary Fig. 2a). A total of 168 samples (84 tumors and 84 nearby tissues) were measured and the results showed good consistency in proteome identification and quantification (Supplementary Fig. 2b). We analyzed the 1008 (168 × 6) raw files together for uniformed quality control and protein identification with 1% global protein false discovery rate (FDR), which resulted in the identification of 11,340 gene products (GPs) (Supplementary Fig. 2c-e). To further increase reliability, we selected 9186 GPs that were detected with at least 2 unique peptides with 1% FDR at the peptide level and with Mascot ion score greater than 20 for subsequent analysis (Fig. 1a, b; Supplementary Data 2). We picked common proteins that were detected in at least one-sixth of the samples (28 cases), resulting in 5439 GPs for molecular subtyping. Proteome quantification was performed as previously reported with the iBAQ algorithm[18] followed by normalization to fraction of total (FOT). We then considered the fact that quantification was more reliable for abundant proteins. The coefficients of variation (CVs) and interquartile range (IQR) of proteins with the FOT over $10^{-5}$ were drastically decreased, suggesting that FOT $> 10^{-5}$ is a good cut-off for accurate quantification (Supplementary Fig. 3a and 3b). Applying this cut-off value, we used the remaining 3619 GPs for principle component analysis (PCA) analysis (Supplementary Fig. 3c). Finally, we selected the 2538 GPs that were differentially expressed over threefold between T and N in at least 10% of the cases and submitted them for clustering and subtyping classification.

Our data showed that the number of GPs identified in the DGC proteome increased steadily until 25 pairs of samples, and approached a plateau between 50 and 84 pairs of samples, suggesting that 84 pairs of samples was sufficient to obtain a detailed DGC proteomic landscape (Fig. 1b). As shown in Fig. 1c, a large number of proteins were annotated as extracellular matrix or in extracellular space, suggesting that our DGC proteome dataset includes the tumor microenvironment.

To obtain genetic background of the samples, we carried out targeted exome sequencing with a panel of 274 cancer driver and GC "hotspot" genes[8, 10, 19] The sequencing resulted in a mean coverage of 234-fold. A total of 7197 somatic variants of 183 genes were detected with mutations in at least one case, and 39 genes were detected with mutations in 5% or more patients (Supplementary Fig. 1d; Supplementary Data 2). TP53, CDH1, KMT2D, RHOA, ARID1A, APC, and PIK3CA were detected as high-frequency mutations (10–46%) (Fig. 1d and Supplementary Fig. 5a), consistent with previous reports[8]. Notably, mutations in the development pathways such as WNT (APC and CTNNB1) and NOTCH (NOTCH1 and NOTCH2) were detected with higher frequencies at 16 and 11%, respectively (Supplementary Data 2).

**General features and altered pathways in the DGC proteome.** Among the 9186 GPs in the DGC proteome, 7443 GPs were found in both tumors and nearby tissues; 1482 GPs and 261 GPs were detected only in tumors or nearby tissues, respectively (Supplementary Fig. 3d). PCA demonstrated a clear distinction between the proteomes of tumors and nearby tissues, revealing an altered proteomic landscape of DGC (Supplementary Fig. 3c). A SAM (significance analysis of microarray) analysis identified 1637

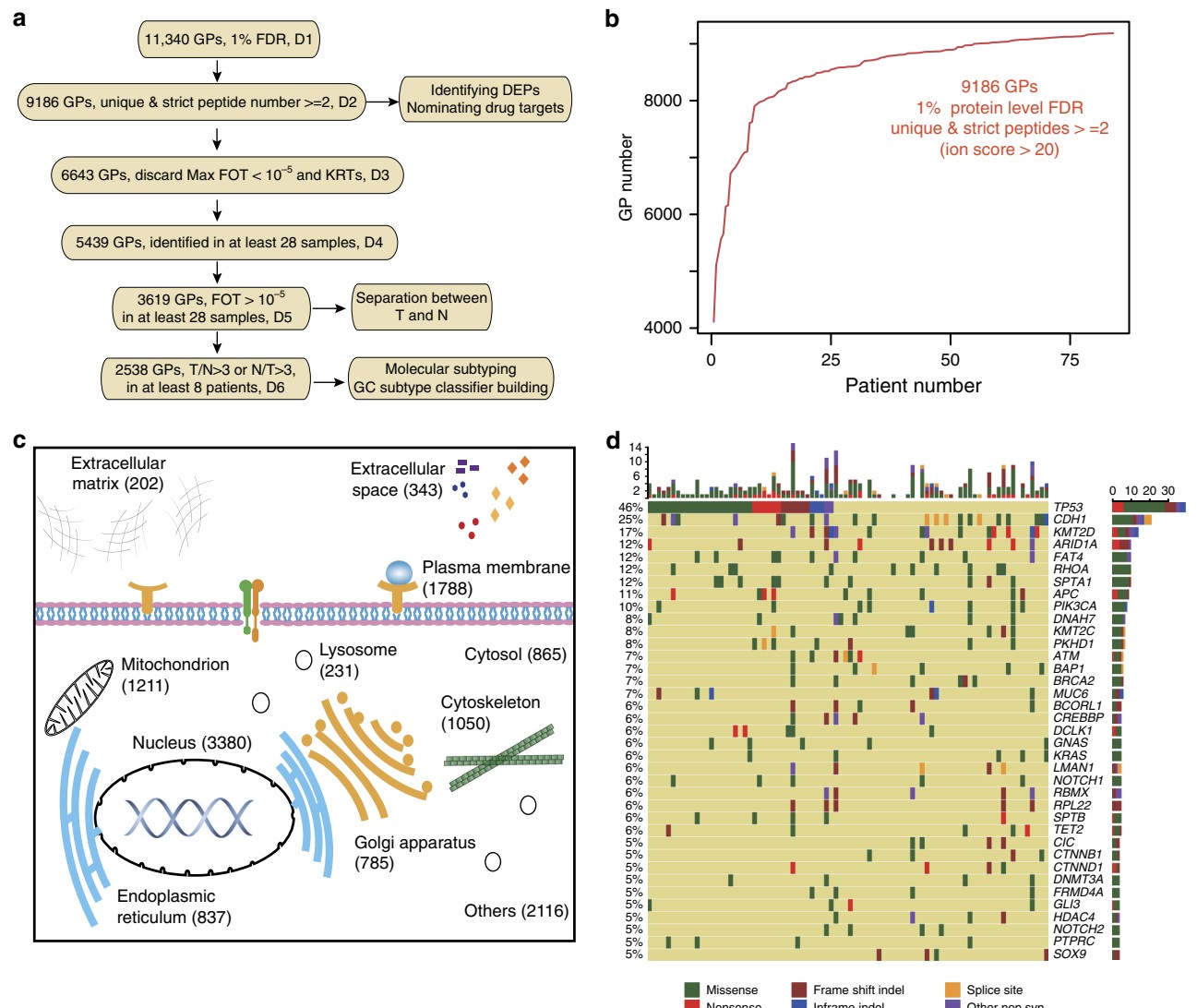

**Fig. 1** A summary of proteomic and genomic analysis of diffuse-type gastric cancer. **a** Proteomic datasets filtered at different levels for various statistical analyses. D1: all gene products (GPs) identified in 168 samples (84 patients) at 1% protein level FDR; D2: high confidence unique proteins identified with ≥2 unique peptides with ion score >20 and peptide FDR <1% that were used for identifying differentially expressed proteins (DEPs) between tumors and nearby tissues; D3: GPs in the medium to high abundance range (FOT >$10^{-5}$ in at least one case), D4: GPs identified in more than one-sixth of the samples; D5: common GPs in the medium to high abundance range (FOT >$10^{-5}$) in D4, D6, differentially expressed proteins. **b** Cumulative number of proteins identified as a function of patient numbers. **c** Subcellular distribution of DGC proteins annotated with Gene Ontology. **d** Targeted exome sequencing was performed. Genes with non-silent variants in at least 4 patients (5%) were depicted on the OncoPrint. Bars on top and to the right of the graph show the number of non-synonymous mutations in each patient and gene, respectively

proteins as differentially expressed between T and N with statistical significance (FDR $q$-value < 0.01 by SAM and differential expression ratio >0.5 or <−0.5), including 1184 up-regulated and 453 down-regulated GPs (Supplementary Fig. 3e; Supplementary Data 3). Gene Ontology annotation indicated that the tumor proteome was significantly enriched in epithelial mesenchymal transition (EMT), cell cycle, DNA replication, checkpoint, E2F, p53 signaling, and inflammatory response pathways, whereas the nearby tissue proteome was enriched in metabolism pathways, such as fatty acid metabolism, oxidative phosphorylation, and amino acid metabolism (Fig. 2a). Notably, many gastric markers (ANXA10, VSIG1, CLDN18, CTSE, TFF2, MUC5AC and MUC6) and signature proteins for stomach functions, including digestion, absorption, secretion, and stomach acid generation such as PGC, GIF, GAST, and ATP4A, were lost in tumors (Fig. 2b). Indeed, stomach-specific proteins annotated in the Human Protein Atlas[20] were significantly down-regulated compared to proteins that were not stomach specific (Fig. 2c), revealing that loss of stomach tissue identity is an important overall feature of DGC.

We then set stricter conditions as the following to define up/down-regulated proteins to seek for commonality of DGC: (1) T/N ratio >3 or <1/3; (2) observed in >75% of the patients; (3) up/down-regulated in >60% of the detected cases. As a result, 490 GPs were identified as differentially expressed proteins in DGC, with 272 up-regulated and 218 down-regulated. Proteins involved in cell cycle regulation (CDK1, HAT1, DNMT1), DNA replication (MCM2-7 helicase, RRM1-2), the condensin complex (SMC2/4, NCAPD2, NCAPG), and metabolism (NNMT, MAGED2), were up-regulated in the tumors (Fig. 2d). Significantly, many proteins from extracellular matrix were detected as differentially expressed between T and N, suggesting that the tumor microenvironment was a significant component in the

altered DGC proteome. Cancer driver pathways, including transforming growth factor (TGF), WNT, NOTCH and interferon (IFN), were up-regulated in DGC with prominent increase in proteins at ligand and receptor levels (Fig. 2e), suggesting the dysregulation of these important cancer driver pathways from the most up-stream in DGC. Notably, while WNT and NOTCH

pathways were detected with genomics, TGF and IFN pathways were only detected with proteomics.

**DGC proteomics subtypes and overall survival**. We employed consensus clustering[21] to identify DGC subtypes based on differentially expressed proteins (D6, Supplementary Data 2). Three

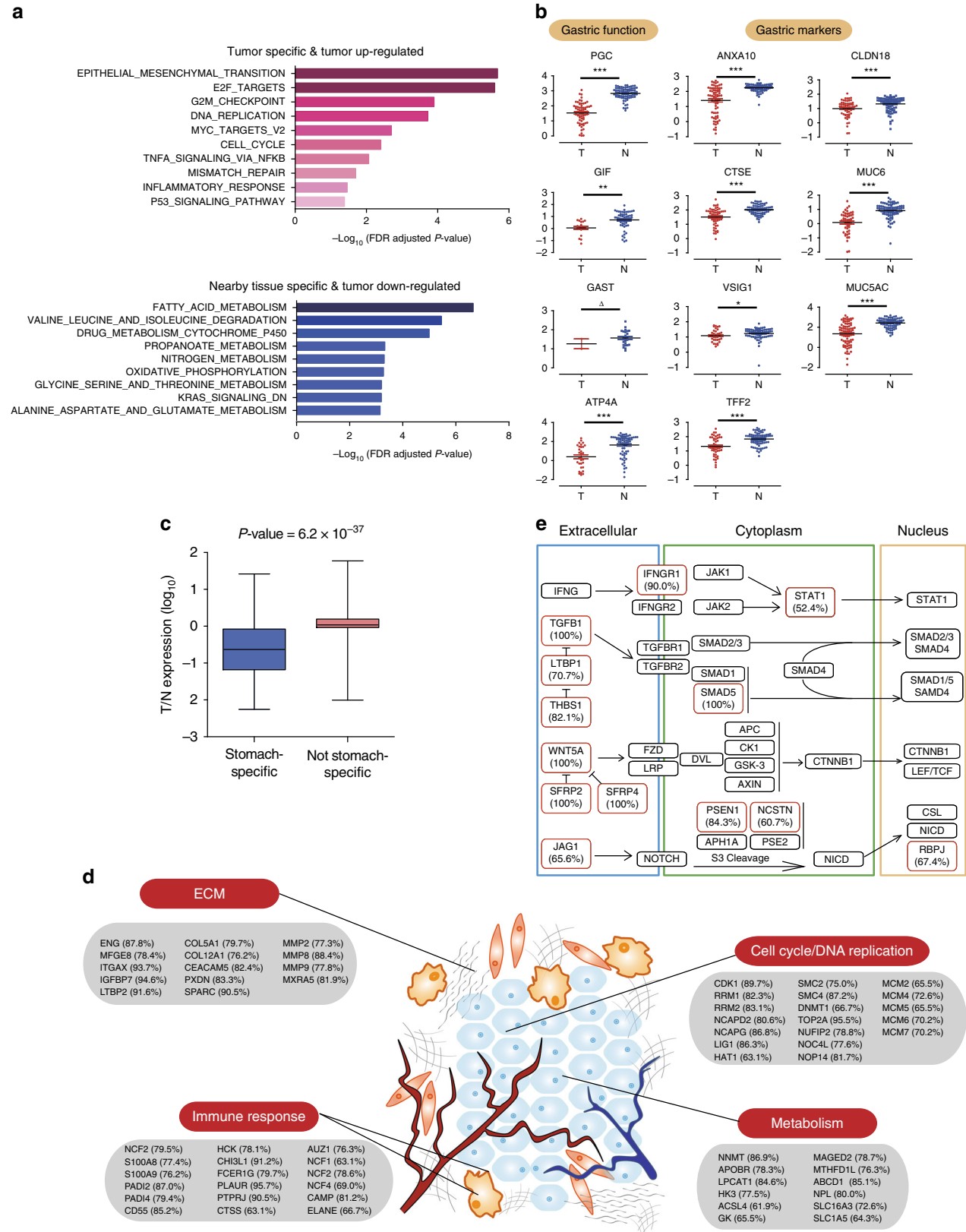

clusters (PX1–3) were evident (Supplementary Fig. 4; Supplementary Data 1): PX1 contained the fewest number of differentially expressed proteins, which were enriched in the cell cycle-related processes (Fig. 3a, b and Supplementary Fig. 4f); PX2 was enriched in not only cell cycle proteins, but also those involved in the EMT process; PX3 was characterized with the enrichment of proteins in the immunological process. Thus, DGC could be subtyped as cell cycle (PX1), EMT (PX2), and immunological process enrichment subtype (PX3) based on their altered proteome patterns alone (Fig. 3c; Supplementary Data 4).

As only one patient of stage I and II met the endpoint (death of cancer), we investigated the overall survival (OS) of 57 patients with clinical stage III or IV with a median follow-up time of 27.2 (8–50) months. The PX1 subtype had the best OS, while PX3 had the worst (log-rank $P = 0.038$, $P$ trend = 0.014, Fig. 3d). Survival analysis of all 82 patients showed similar trend regardless of the TNM (tumor, node, metastasis) stages (Supplementary Fig. 4e). Remarkably, cancer subtyping was associated with clinical outcome based solely on the altered cancer proteome, irrespective of genetic background. Furthermore, proteomics subtyping could also serve as an independent predictive factor (Cox $P$ trend = 0.006, hazard ratio (HR) = 2.757) in the multivariable analysis after adjusting for clinical stage and other covariates (Table 1 and Supplementary Table 1). We found no statistically significant prognosis improvement by chemotherapy for the PX3 group (log-rank $P = 0.831$, Fig. 3d); majority of the PX1 and PX2 patients received adjuvant chemotherapy, preventing a sound statistical assessment of the chemotherapy response.

**Correlation of tumor proteomes with DNA mutations**. To investigate the correlation between genome and proteome, we first investigated DNA mutations of driver genes with proteome alteration patterns. Among the three subtypes, PX1 has the fewest number of DNA mutations, while PX3 has the most number of mutations (Fig. 4a and Supplementary Fig. 5b). DNA mutations in PX3 showed enrichment in several pathways, including the CXCR4, PI3K-AKT, and focal adhesion pathways (Fig. 4a; Supplementary Data 5). Next, we examined the connection between gene mutation and alteration in protein abundance. Of the 183 mutated genes, only 9 gene products exhibited altered protein expressions compared with the wild type when all nonsynonymous mutations were taken into consideration; of the 87 mutated genes with truncating mutation, only 4 genes (RBMX, ARID1A, SOX9, and TMPO) exhibited significantly lower protein expression when considering only truncating mutations (Supplementary Fig. 5c and 5d; Supplementary Data 5), suggesting that very few mutations strongly impact protein expression. Of the 9 genes that did show altered protein expressions, mutant tumor suppressor gene (TSG) products (ARID1A, ATM, BAX, PLEKHA6, SOX9) exhibited reduced expression, whereas the oncogene product (MED12) exhibited increased expression, as expected. One exception is the tumor suppressor NF1, whose mutation was correlated with higher protein abundance.

A considerable number of driver genes (>13%, 25/183), such as AR, MYC, PPARA, and FBXW7, were detected with DNA mutations but their proteins were not detected in tumors or nearby tissues (Supplementary Fig. 5e, Supplementary Data 2). Notably, 13% (24/183) of the mutated genes, including KRAS, NRAS, CDH1, ATM, and PTEN, were detected only in nearby tissues but not in tumors (Fig. 4b). The latter examples excluded the trivial explanation that the mutant proteins were not amenable for detection with MS. As the DGC proteome covered more than 9000 gene products in our study, the undetected proteins with detectable DNA mutations were probably among the extremely low abundant proteins that are either below the detection limit of the current method or not expressed at all in the tumor. While loss of TSG proteins in tumors was expected, failure in detection of oncoproteins in tumors raised a serious concern about using DNA sequencing alone to "predict" protein expression and highlighted the absolute necessity of measuring proteins directly for precision medicine.

We also noted that up- and down-regulation of oncogene and TSG were not always consistent with their predicted roles in carcinogenesis among the well-annotated 138 cancer driver genes[19] (Fig. 4c). For example, TSG products RB1 and NF1 were often increased, while the oncogene products KRAS and CTNNB1 were decreased in tumors. These observations were consistent with the immunohistochemistry (IHC) results in the Human Protein Atlas[20]. We suspect that compensation could be one contributing factor to account for such a behavior, in which some TSG pathways are hyper-activated and oncogene pathways are tuned down to maintain tumor homeostasis.

Next we investigated how a single gene mutation correlates with the alteration of the cancer proteome, namely alterations of other proteins and pathways. For genes with DNA mutation frequency greater than 10% (TP53, CDH1, KMT2D, ARID1A, FAT4, RHOA, SPTA1, APC, and PIK3CA), we mined the data using Wilcoxon rank-sum test and found significantly altered proteins between samples with and without the mutation (Supplementary Data 5). In total, the 9 most frequently mutated genes were significantly correlated with the alteration of expressions of 3958 proteins in tumors, ranging from 126 in the CDH1 mutants to 880 proteins in the PIK3CA mutants (Supplementary Fig. 5f and Supplementary Data 5).

We chose CDH1 mutations as an example to elucidate how gene mutations were associated with cancer pathway alterations. In total, 21 and 62 samples were found with or without CDH1 mutations, respectively. The Wilcox test identified 87 up-regulated and 39 down-regulated proteins associated with CDH1 mutations (Supplementary Data 5). Among them, proteins in the canonical cancer driver pathways, including WNT5A, epidermal growth factor receptor (EGFR), and AKT2, were up-regulated in tumors with CDH1 mutations, while the WNT inhibitor, LRP10, was down-regulated, suggesting that up-regulations of WNT, EGFR, and AKT pathways are operational in CDH1-mutated tumors (Fig. 4d). Similarly, tumor suppressor proteins involved in adhesion, including EPHB3 and IGSF8, which were reported to suppress cancer metastasis and TGF-β signaling[22, 23], respectively, were significantly down-regulated, suggesting that EPHB3 and IGSF8 may function downstream of CDH1 to mediate cell–cell adhesion. Interestingly, FOCAD, a protein involved in focal adhesion, was also up-regulated in the

**Fig. 2** Proteomic features of the diffuse-type gastric cancer. **a** Top ranked pathways that are significantly altered in tumors as compared with nearby tissues. **b** Significantly decreased gastric function proteins and gastric mucosa signature proteins in tumors and nearby tissues. The y-axis represents $\log_{10}$ (FOT) +5. *$P < 0.05$, **$P < 0.01$, *** $P < 0.001$ (Wilcoxon rank-sum test), $^{\Delta}$Insufficient sample size. **c** Box plots of $\log_{10}$ transformed T/N ratios of stomach-specific and not stomach-specific proteins; whiskers show the 1.5-fold IQR, $P$-value was calculated by Wilcoxon rank-sum test. **d** Common proteins up-regulated in majority of tumors that were classified in the four functional categories. (% in parenthesis denotes percentage of tumor samples that exhibit >3-fold change in 84 patients). **e** Altered proteins in the WNT, NOTCH, TGF, and INF pathways. Differentially expressed proteins in tumors (T/N >3) were boxed with red color. The number in parenthesis is the percentage of samples with overexpression when the protein was detected

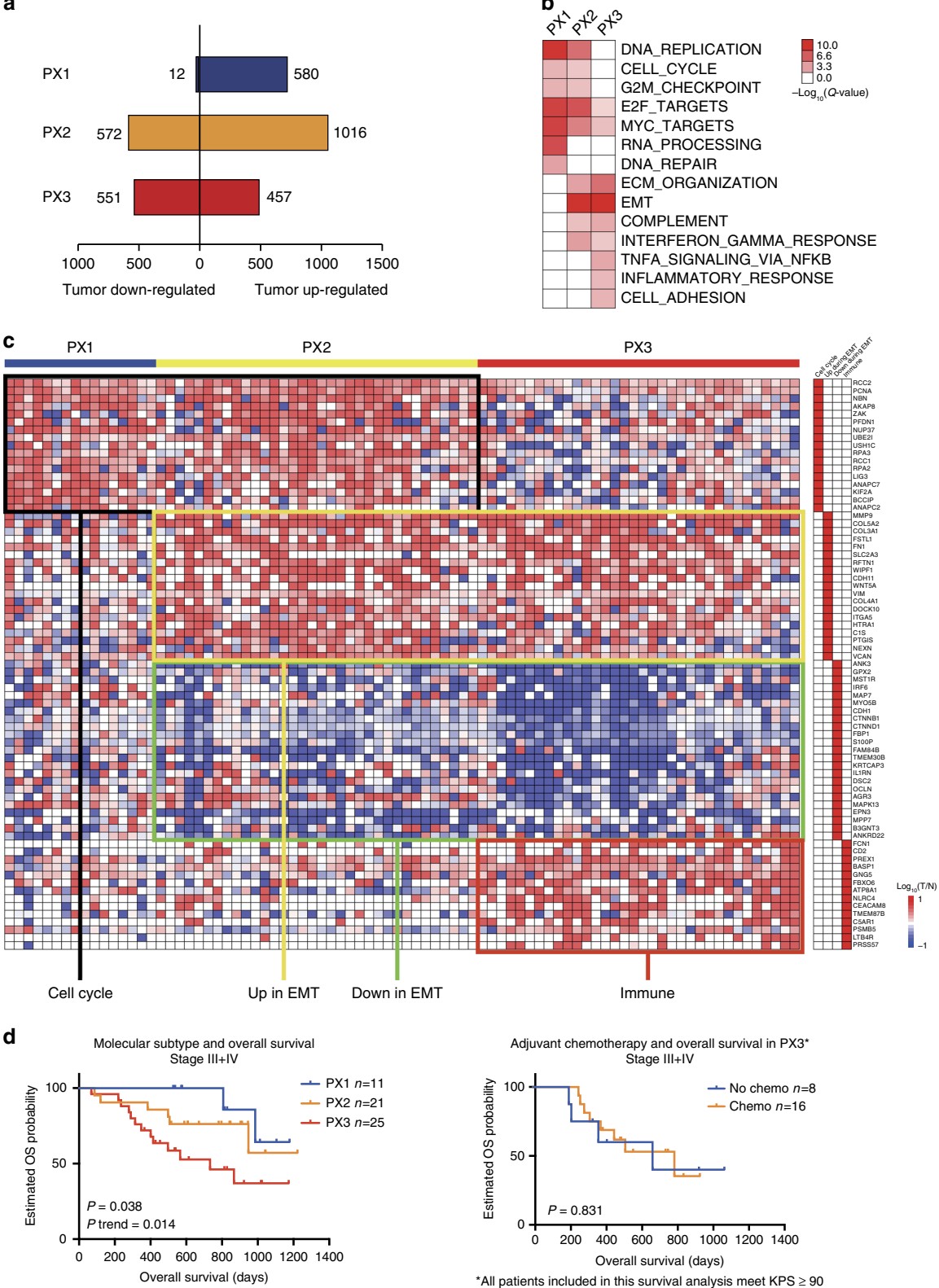

**Fig. 3** Molecular subtyping of DGC based on altered proteomes and their correlations with clinical outcomes. **a** The number of differentially expressed proteins in each cluster. **b** Major signaling pathways enriched in PX1, PX2, and PX3. **c** A heat map of selected proteins representing major altered signaling pathways across 84 patients in PX1–3. **d** Left: the association of molecular subtypes with overall survival of stage III and IV patients (Kaplan–Meier analysis, *P*-value from log-rank test); Right: the association of adjuvant chemotherapy with overall survival of stage III and IV patients in PX3 (Kaplan–Meier analysis, *P*-value from log-rank test)

**Table 1 Univariate and multivariate analysis of overall survival in 82 patients**

| Variable (n) | Univariate analysis | | Multivariate analysis | |
|---|---|---|---|---|
| | HR (95% CI) | *P*-value | HR (95% CI) | *P*-value |
| Age[a] | 1.037 (1.001–1.076) | **0.045** | 1.033 (0.994–1.074) | 0.102 |
| Gender | | | | |
| Male (51) | 1.0 | | | |
| Female (31) | 1.122 (0.478–2.638) | 0.791 | | |
| Adjuvant chemotherapy | | | | |
| Without (21) | 1.0 | | 1.0 | |
| With[b] (61) | **0.330 (0.138–0.788)** | **0.009** | 0.834 (0.282–2.466) | 0.742 |
| Tumor site | | | | |
| Cardia, GEJ (20) | 1.0 | | | |
| Body (33) | 1.405 (0.439–4.501) | 0.567 | | |
| Antrum (29) | 1.497 (0.450–4.986) | 0.511 | | |
| Clinical stage[a] | | | | |
| (Ib to IV) | **7.065 (2.661–18.760)** | **<0.001** | **17.91 (3.529–90.91)** | **<0.001** |
| TP53 mutation | | | | |
| Wild type (45) | 1.0 | | | |
| Mutant (37) | 1.407 (0.608–3.260) | 0.423 | | |
| Profiling cluster | | | | |
| PX1 (17) | 1.0 | | 1.0 | |
| PX2 (32) | 2.186 (0.453–10.560) | 0.330 | 4.280 (0.731–25.052) | 0.107 |
| PX3 (34) | 4.221 (0.950–18.760) | 0.058 | **9.785 (1.597–59.935)** | **0.014** |
| *P* trend | **2.009 (1.070–3.772)** | **0.025** | **2.757 (1.329–5.722)** | **0.006** |

GEJ, gastroesophageal junction; HR, hazard ratio; CI, confidence interval
[a]Continuous variable
[b]Patients proceed through at least one cycle of adjuvant chemotherapy. Significant data are emphasized in bold

mutants, suggesting a possible compensation mechanism when cell–cell adhesion was compromised with loss of CDH1 function. This analysis provided new protein candidates to investigate the *CDH1*-mediated pathways. The up-regulated protein ASPN has been reported to reside in the tumor stroma and promote co-invasion of cancer-associated fibroblasts and cancer cells[24]. This clue indicates that *CDH1* mutation influences not only the tumor cells, but also the tumor microenvironment. Thus, the *CDH1* mutations appeared to be associated with three cancer-related processes—the canonical cancer driver pathway, cell–cell adhesion, and the tumor microenvironment (Fig. 4d and Supplementary Data 5).

For the 9 genes with higher mutation frequency, we calculated the Jaccard index between any two altered proteomes caused by the mutations to evaluate how similar they were (Supplementary Data 5). Only *APC* and *PIK3CA* mutations had weak similarity in the altered proteome (Supplementary Fig. 5g) suggesting their coordinated regulation, which has been show previously in colorectal cancer[25].

**Nomination of druggable candidates for DGC.** We first evaluated the 159 druggable proteins elected previously through comprehensive analysis of genomic data of different cancers, which we referred to as "genomic targets"[26–28] (Supplementary Data 6). Our proteomic analysis had confidently identified 103 of them (detected in more than 5 patients). Here, we limit the druggable targets to those that are over-activated (or over-expressed) in tumors. In our dataset, 18 of them met this condition and, among them, growth regulators such as PDGFR, PI3KCA, AKT3, and EPHB2, as well as cell cycle regulators, including CDK4, ATR and AURKA, could be potential drug targets. When we used Cox regression analysis to identify prognosis unfavorable therapeutic targets whose higher protein expression levels are associated with worse OS, none of the above genomic targets could pass the test; instead, two candidates, EPHB2 and TP53, were associated with better OS when

overexpressed (Fig. 5a and Supplementary Fig. 6a; Supplementary Data 6).

We then sought to nominate drug target candidates based on proteomics data. We applied the following criteria in the selection: (1) they have to be differentially overexpressed in more than 50% of the tumors; (2) they belong to functional categories that are conventionally considered as druggable, namely enzymes, GPCR (G protein coupled-receptor), kinases, ion channels, transmembrane and other membrane proteins, and extracellular membrane proteins, against which neutralizing antibodies could be developed; (3) their overexpression is correlated with poor survival (Cox regression, log-rank test *P*-value < 0.05, HR > 1). We identified 23 potential proteomic drug candidates that met all these conditions (Fig. 5b, c; Supplementary Data 6).

These 23 proteomic drug candidates, which reside in tumor cells, extracellular matrix, or immune cells, suggest that four categories of DGC vulnerability may be exploited (Fig. 5d): (1) the canonical cancer growth pathway—JAG1 and IGFBP3 that are components of the NOTCH and insulin-like growth factor (IGF) pathways, (2) metabolism and oxidative stress—GZMK, CYBA, SLC1A5, SLC16A3, PGM2L1, and SCD, which indicate the dysfunction in reactive oxygen species (ROS), oxidative stress, and metabolic pathways for glutamine, lactate, and unsaturated fatty acid, (3) cell adhesion and invasion—PLSCR1, CLDN1, MMP8, ITGB2, and GPRC5B, and (4) immune-modulation in tumors and tumor microenvironment—IDO1, PTGS2, CD55, CD97, HLA-DQA1, SPN, CD300A, EMR2, SIPRA, and UNC93B1, which are present in different immune cell types, including antigen-presenting cells (APCs), T cells, natural killer cells (NK cells), and macrophages.

The first vulnerability of DGC is the canonical cancer growth pathways, including NOTCH and IGF pathways. JAG1 is the ligand in the canonical Notch pathway in tumor growth through maintaining cancer stem cell populations, promoting cell survival, inhibiting apoptosis, and driving cell proliferation and metastasis[29]. IGFBP3, which is a key regulatory molecule in the IGF axis and could be either tumor suppressor or promoter in

different tumor types, was reported to be highly expressed in aggressive breast cancer, advanced stage of melanoma, and higher in metastatic than non-metastatic tumors in pancreatic endocrine neoplasms[30].

The second vulnerability is the metabolism and oxidative stress. One of the hallmarks in the advancement of cancer cells is their ability to overcome and acquire resistance to adverse growth conditions, including ROS and aberrant metabolism. CYBA (also known as p22phox) binds to and stimulates NADPH oxidase (NOX) to produce excessive amounts of ROS, which can cause oxidative damage to lipids, proteins, and DNA, making the cellular environment unfavorable for normal cells to grow but adaptable for tumor cells to survive[31]. Outside of the cancer cell, the overexpressed GZMK (Granzyme K) in the extracellular

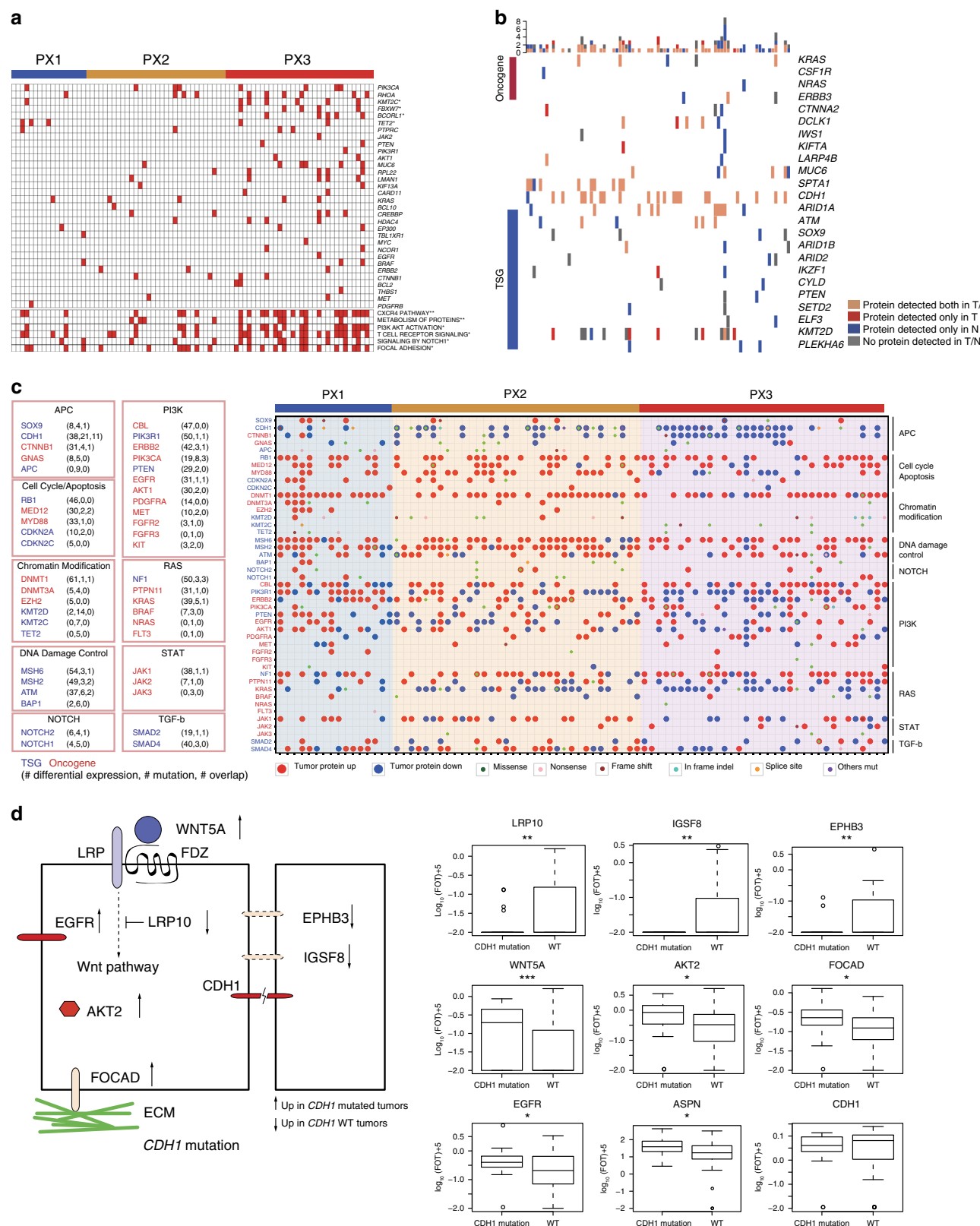

matrix could induce rapid ROS generation and the collapse of the mitochondrial inner membrane potential to provide double jeopardy by oxidative stress[32]. Overexpression of SLC1A5 (ASCT2), SLC16A3 (MCT4), PGM2L1, and SCD suggests that DGCs have modified their metabolic pathways for glutamine[33], lactate[34], and unsaturated fatty acid[35].

The third vulnerability is the cell adhesion and invasion. PLSCR1, a transmembrane lipid transporter, is involved in rapid $Ca^{2+}$-dependent trans-bilayer redistribution of plasma membrane phospholipids and is highly expressed in many tumor tissues[36]. MMP8, ITGB2, and CLDN1 are well-studied proteins that have evolved with tumor cell's higher ability for invasion. In a clinical trial, an antibody against CLDN18.2, when added to standard chemotherapy, resulted in a 53% reduced risk for progression and a 49% reduced risk of death for advanced gastric cancer patients (FAST, NCT01630083, Germany)[37]. Our results suggest that a therapy targeting tight junction for DGC could be expanded to CLDN1.

The fourth vulnerability is the immune-modulation in the tumor microenvironment. IDO1 has been reported to be overexpressed in many tumors, in either the tumor cells themselves or tumor-associated cells such as dendritic cells (DCs), macrophages, and endothelial cells. Overexpression of IDO1 increased the accumulation of kynurenine (Kyn), a metabolite that inhibits T-cell function and biased DCs and macrophages toward an immunosuppressive phenotype[38]. Another enzyme that plays a vital role in immune suppression is PTGS2 (COX-2), which produces the arachidonic acid prostaglandin E2 (PGE2) that selectively suppresses effector functions of several immune cells and also alters cytokine expression profiles of DCs to suppress antitumor cytotoxic T cells[39, 40]. Other candidates, including CD55, CD97, HLA-DQA1, and CD300A, are distributed in many types of immune cells (APC, T cell, NK cell, and so on.) and correlated with poor prognosis of a variety of cancers[41–44]. A list of drug candidates similarly identified that were subtypes specific (PX1, 2 and 3 specific) can be found in Supplementary Fig. 6b.

**The protein landscape of cancer immunotherapy in DGC.** Recent cancer immunotherapy has revolutionized the treatment of cancers. The PX3 subtype has the worst prognosis and is resistant to chemotherapy. The enrichment of immune-related proteins raised the hope for cancer immunotherapy as a treatment option.

Surface receptors of immune cells mediate intracellular and cell–cell communications to active effector cells to kill tumor cells. This process can be hijacked and diverted by tumor cells to evade immune killing. We first surveyed the expressions of receptor–ligand pairs that are involved in forming the immune synapses, as described in published reviews[45, 46]. Ten pairs of receptor–ligands were identified in the DGC proteome and 9 of them were overexpressed (T/N > 3-fold) in tumors, ranging from 1.2% to 34.5 % of the samples (Fig. 6a), indicating the presence of considerable number of immune cells in the DGC samples. Majority of the receptor–ligands that were detected likely originate from the APC side, while only one receptor (CD27) is from the T-cell side in the APC–T-cell immune synapses.

We next examined protein expression of drug targets that are currently being developed for cancer immunotherapy[47]. We detected 19 targets that were expressed at different levels (Fig. 6b). PD-L1 (CD274) was only detected in two patients and expression levels were extremely low. Notably, expressions of TMEM173 (STING), ARG1, NT5E, CD40, IDO1, SIRPA, CD276, and FCGR1A were high in tumors and were highly enriched in the PX3 group. An overview of potential immunotherapy targets and related proteins can be found in Supplementary Fig. 6c[45–47].

IDO1 and ARG1 are two immune suppressive mediators with inhibitory agents in clinical trials. We performed IHC of IDO1 on selected patients' samples that displayed differential IDO1 expression by proteomic analysis. The high IHC scoring was consistent with the high protein abundance measured with mass spectrometry (FOT) and western blot (Fig. 6c). We defined samples as IDO1 high or ARG1 high if the abundance of these proteins is in the upper quartile (75th percentile) of all samples. IDO1-high, ARG1-high, and particularly both high cases were significantly enriched in the PX3 group, while other parameters that may have predictive values for responders of immunotherapy, such as mutation numbers (MSI-high), EBV infection, and intratumoral tumor-infiltration lymphocytes (IT-TILs), failed to stratify patients into the three subtypes (Fig. 6c). We did IHC of T-cell marker CD8+ and hematoxylin and eosin (H&E) staining to calculate the percentage of IT-TILs. The IT-TIL values showed no difference among the three groups, but the CD8+ T cell-high patients were 2/16, 5/34, and 11/34 in PX1, PX2, and PX3, respectively, confirming a heightened cellular immune response in the PX3 group (Fig. 6c, d and Supplementary Fig. 6d). Taken together, these data verified that the PX3 group has a more active immune response and could be the prime candidate for cancer immunotherapy.

## Discussion

We presented a proteomic landscape of DGC with 84 pairs of tumors and matching nearby tissues. This work is a logical extension of the TCGA-affiliated CPTAC project, which has carried out proteogenomic analyses of colorectal, breast, and ovarian tumors with extensive genome, transcriptome, and proteome profiling but without information from the matching nearby tissues[14–16]. Our work provides a direct comparison of these information, presenting a panoramic view of the altered cancer proteome and allows the analysis and extraction of altered signaling pathways in DGC at the proteome level. An independent cohort would be required to validate the clinical relevance and subtypes we observed.

Molecular subtyping of cancers, aimed to stratify patients into subtypes associated with clinical outcomes, therapy responses, and biological characteristics, has been a long sought-after goal of mapping the genetic landscape of cancer. DGC is mainly classified as the genome stable subtype in TCGA and accounts for the majority of the MSS/EMT subtype in ACRG with the worst prognosis[8, 9]. Our proteomic analysis has further separated DGC

**Fig. 4** Correlation of genomic mutations and protein expressions. **a** Differentially mutated genes and their pathways in PX1–3, *P < 0.05, **P < 0.01 (Fisher's exact test). **b** Protein expression status of selected mutated genes. **c** Differential protein expression of mutated genes in the major cancer driver pathways. The left panel summarizes the correlation of altered protein expression and gene mutations of selected proteins in major oncogenic and tumor suppressive pathways; numbers in parenthesis represent number of patients that differential protein expression were detected, number of mutations detected, and number of cases where both were detected, respectively. Right panel shows differential protein expression and gene mutations in each patient. Large dots depict changes in protein expression, and small dots depict a variety of gene mutations. **d** Altered expression of tumor proteins associated with *CDH1* mutations. Boxplots show protein expression levels of *CDH1*-mutated and *CDH1*-wild-type patients (*P < 0.05; **P < 0.01; ***P < 0.001, Wilcoxon rank-sum test). Whiskers show the 1.5-fold IQR

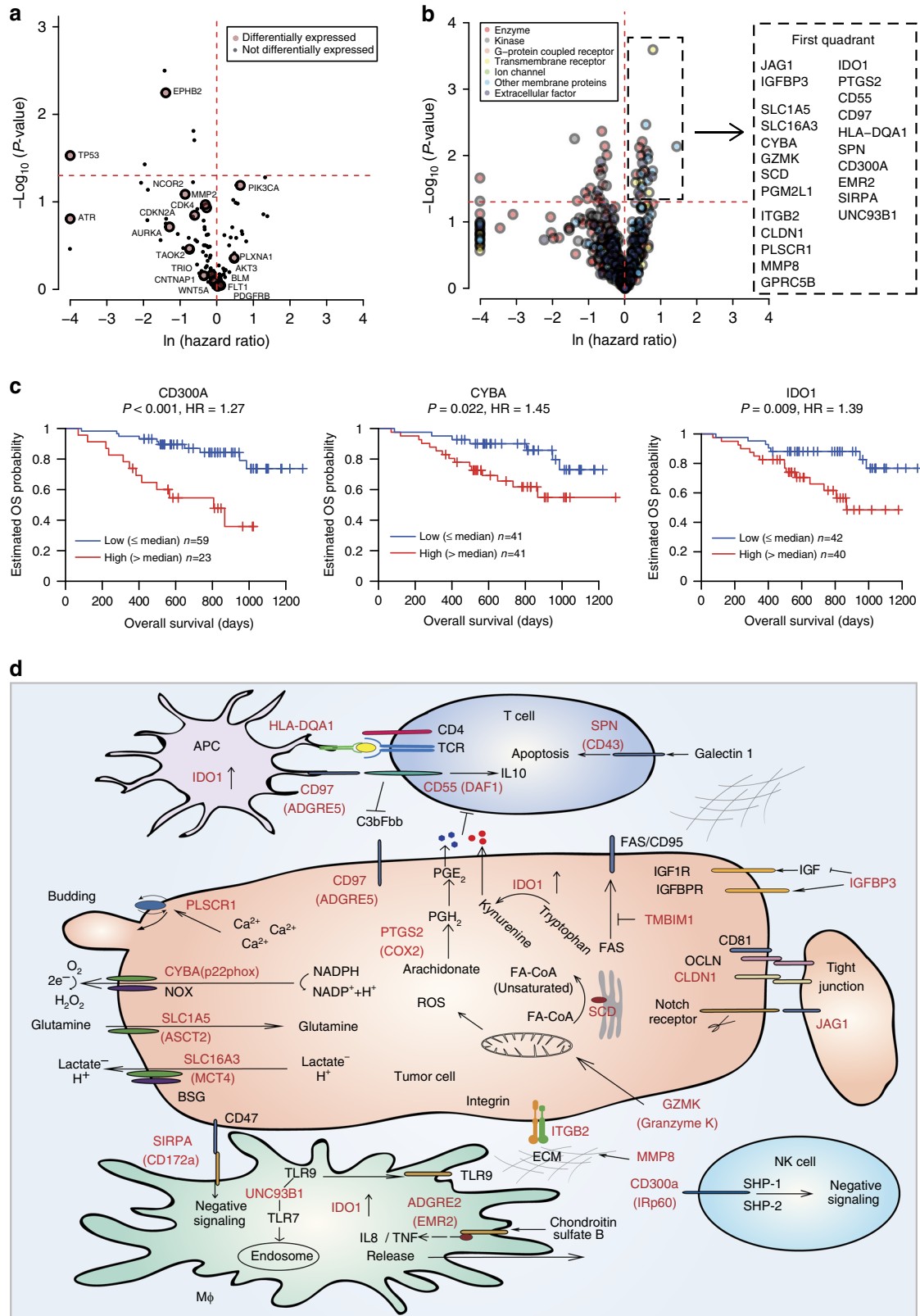

**Fig. 5** Nominating potential druggable proteins for DGC. **a** The association between protein expression (T/N ratio) and overall survival of genomic data discovered drug targets. The ln (Hazard ratio) (*x*-axis) and log$_{10}$(*P*-value) (*y*-axis) were calculated from Cox proportional hazards regression analysis. Ln (Hazard ratio) >4 or <−4 were plotted with 4 or −4. Large dots depict proteins overexpressed in tumors, and small dots depict proteins that are not overexpressed in tumors. **b** The association between protein expression (T/N ratio) and overall survival of proteomic data discovered drug targets. A list of 23 proteins overexpressed in tumors with log-rank *P*-value < 0.05 and ln (hazard ratio) >0 is included in the box on the right. **c** The association between protein expression of CD300A, CYBA and IDO1 and overall survival (Kaplan–Meier analysis, *P*-value from log-rank test, high means T/N >median value). **d** Cellular localization and signaling pathways of nominated drug target candidates. The nominated druggable protein candidates are depicted in red

into three subtypes, revealing the heterogeneity and diversity of DGC at the proteome level. The proteomic subtypes feature distinctly altered signaling pathways and clinical outcomes. Our analyses indicate that the PX3 subtype may not benefit from chemotherapy, but could be the prime target for immunotherapy.

How to translate proteomic subtyping into clinical application will be an important research direction in the future.

Notably, while a considerable number of genes were identified with DNA mutations, their gene products were never detected in the DGC proteome, and several oncogene products were not detected in the tumors but were detected in the nearby tissues.

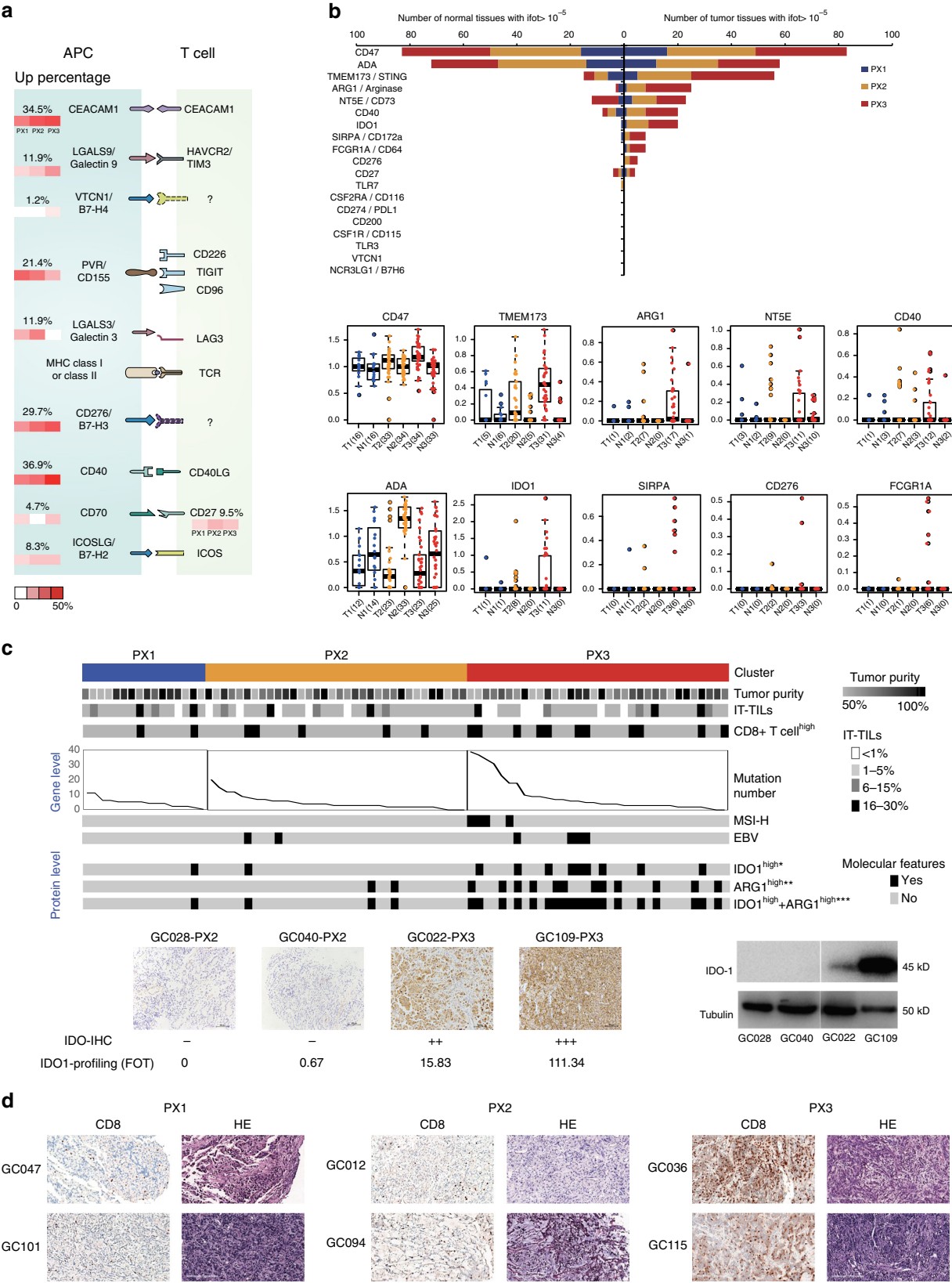

Similar observations have also been made in the previous CPTAC studies, as many amplified genes were not transcribed into mRNA or translated into proteins[14]. It appears that DNA mutations can be viewed as historical records of what has happened during the evolution of the tumor, some of which may no longer be functioning at the time of surgical resection, whereas protein analysis portrays the current states of the tumor. Our observations, especially the loss of activating oncogene products in tumor samples, add a cautionary note on nominating treatment candidates based solely on DNA mutations and further strengthen the necessity of measuring proteins in precision medicine. As our ability to measure proteins continues to improve, it could be expected that protein measurements would become an integral component for precision medicine.

The targeted DNA exome sequencing revealed a poor correlation between genomic and proteomic data, as has been demonstrated in the CPTAC projects[14, 16], using more extensive analyses of the genome and transcriptome. The discordance of gene mutation and protein abundance could be explained by (1) tumor heterogeneity, in which DNA mutations in a small number of tumor cells were detected, while their alteration in protein abundance is masked by neighboring cells that were abundant and genetically wild type; (2) limited number of samples with DNA mutations, which diminishes the power of statistical analysis to calculate the correlation. For higher frequency mutations, such as CDH1, ARID1A, and RHOA, proteome alteration information can indeed be extracted. For example, the correlation of CDH1 mutations to the activation of WNT, EGFR, and AKT pathways as well as the inactivation of tumor suppressors, such as EPHB3 and IGSF8, provides several hypotheses to test.

Analyzing proteomes of both tumor and the nearby tissue from the same patient also allowed us to more reliably nominate potential drug targets. We suggest that nominating drug targets based on their overexpression in tumor and druggability is not always adequate. For instance, we found that, while many proteins belong to the druggable categories, including several "star" molecules including PDGFRB, CDK4, and PI3KCA derived from cancer driver gene analyses, their elevated expressions are not correlated with poor survival. In fact, almost all druggable proteins derived from the cancer driver genes do not exhibit positive correlations between high expression and poor prognosis, making their targeted inhibition a questionable treatment strategy. We propose to consider the correlation between overexpression and OS in addition to differential expression in the tumor when nominating drug target/candidates for treatment and developing drugs. It will be interesting to investigate in the future the 4 vulnerabilities and the 23 potential drug candidates that we nominated for DGC.

Our analyses of tumors with their microenvironments also painted an immune protein landscape in DGC. Tumor-infiltrating immune cells, for example, CD8+/CD3 T cells, are associated with good prognosis in ovarian, colon, breast cancer, and other solid tumor types[48, 49]. However, this does not seem to be the case in gastric cancer. It was reported by Thompson et al.[50] that an increased CD8 infiltration is correlated with impaired progression-free survival and OS in gastric cancer and gastro-esophageal junction cancers, and patients with higher CD8+ T-cell densities also have higher PD-L1 expression. The reason for such difference is not clear, but it may indicate an adaptive immune resistance mechanism. Compared with an overall response rate (ORR) of 55% in non-small-cell lung cancer (clinical trial KEYNOTE-021)[51], an anti-PD-L1 therapy only received an ORR of 11.2% in GC (clinical trial KEYNOTE-059)[52]. Considering the low and infrequent expression of PD-L1 (CD274) in this study, several other immune checkpoint blockers, especially the ones identified in the PX3 group, deserve further detailed investigation and development as potential targets for DGC.

In summary, our study provides a rich resource for determining the major pathway alterations in DGC, and demonstrates the advantage of understanding cancer in the context of tumor microenvironment at the proteome level. How to translate these data and information into clinical practice and care will be the future direction for cancer research.

## Methods

**Biospecimen collection and pathology and clinical data**. We screened 2451 gastric cancer patients who took total or subtotal gastrectomy at Beijing Cancer Hospital, Beijing, China (from December 2012 to July 2015), and selected 146 cases of DGC. Among the excluded 2305 patients, 428 were treated with neoadjuvant chemotherapy or chemo-radiation therapy before operation, 92 were diagnosed with gastrointestinal stromal tumors, 751 were intestinal-type gastric cancer, 644 were mixed-type gastric cancer, and 390 did not have enough tumor tissues or nearby gastric tissues. All cases were staged according to the seventh edition of American Joint Committee on Cancer (AJCC) staging system. Each specimen was collected within 30 min after operation, cleaned with sterile towel, immediately transferred into sterile freezing vials and immersed in liquid nitrogen, then stored at −80 °C until use. The protocol was approved by Beijing Cancer Hospital Medical Research Ethics Committee (2015KT70). We collected written informed consent from all participating patients.

Tumors and their nearby tissues were evaluated by pathologists. Nearby tissues were designated as non-cancerous and were greater than 5 cm away from the surgery margin. Each specimen was cut into four pieces under −40 °C. One was formalin fixed and paraffin embedded for pathology examination; one was used for protein profiling, one was used for DNA sequencing, and one was stored for future use. H&E-stained sections were examined by two expert gastrointestinal pathologists (Z.L. and Yumei Lai) independently to confirm: (1) diffuse type (Lauren type); (2) >50% tumor cell nuclei; (3) <20% necrosis in tumor tissue; (4) no tumor cells in nearby tissue (Supplementary Data 1). Mesenchymal percentage, normal cell percentage, signet ring cell proportion, lymphovascular invasion, and nearby tissue status (superficial gastritis, atrophic gastritis, intestinal metaplasia or dysplasia) were also determined. Among the 146 pairs of DGC samples, 56 did not pass the criteria due to less than 50% of tumor cells in tumor tissue, 3 failed because nearby tissue contained tumor cells, and 1 failed because of muscle layer tissue in nearby tissue. An additional two patients were excluded because gastric cancer was diagnosed as second primary tumor after lung cancer. Specimens in dry ice were transferred to Beijing Proteome Research Center within 3 h after surgery. The remaining 84 diffuse type were processed for proteome profiling, and 83 were

---

**Fig. 6** The protein landscape of cancer immunotherapy in DGC. **a** Multiple receptor–ligand pairs found in antigen-presenting cells (APCs) and T cells (figure adapted from ref.[45] and ref.[46]). Shade of the each of the three blocks represents the percentage of patients with the protein overexpressed in PX1–3, respectively, and the number above the block denotes the total percentage of all patients with the protein overexpressed. **b** Expressions of 19 immunotherapy targets in clinical development. The y-axis of the box plots at the bottom represents $\log_{10}$ (FOT ×$10^5$) of each protein; the number in parenthesis in the x-axis represents number of patients with FOT >$10^{-5}$. T1–3 tumor tissues of PX1–3; N1–3 nearby tissues of PX1–3. The box plots show the median, 25th and 75th percentile values (horizontal bar, bottom and top bounds of the box), and whiskers show the 1.5-fold IQR. **c** Comparison of various parameters among three clusters. Patients in PX1 (blue), PX2 (orange), and PX3 (red) are ordered by mutation numbers. *$P < 0.05$, **$P < 0.01$, *** $P < 0.001$ (Chi-square test), IDO immunohistochemistry staining of representative examples (bottom, shown at ×100 magnifications, scale bar: 100 μm), their IHC scores, the corresponding FOT ×$10^5$ values obtained from protein profiling, and IDO1 protein expression measured by western blot are shown. IT-TILs intratumoral tumor-infiltrating lymphocytes, CD8+T-high CD8-positive T-cell number ≥298 per μm$^2$ (see Supplementary Figure 6d for details), MSI-H microsatellite instability high, EBV Epstein–Barr virus status; IDO1/ARG1-high ≥upper quartile of all detected values. **d** H&E and CD8 IHC staining in representative examples in PX1–3, shown at ×100 magnification (scale bar: 200 μm)*$P < 0.05$, **$P < 0.01$, *** $P < 0.001$ (Chi-square test),

processed for targeted sequencing, except for one patient due to low DNA quality. All except 2 patients (GC055 and GC096) received follow-up every 6 months from the date of surgery.

Demographics, histopathologic information, family history of cancer, primary tumor location, first recurrent site, treatment details including chemotherapy drugs, doses and routes of administration, and outcome parameters were collected. The date of operation was used as a surrogate for the date of initial diagnosis. OS was defined as the interval between the date of initial surgical resection to the date of last known contact or death. Disease-free survival was defined as the interval between the date of initial surgical resection to the date of progression or to the last follow-up date. The date of progression was defined as the date of documented recurrences by imaging evidence (computed tomography, magnetic resonance imaging, or positron emission tomography). With or without chemotherapy in this research was defined as with or without at least one cycle of adjuvant chemotherapy.

**Immunohistochemistry staining and evaluation**. IHC for MLH1, PMS2, MSH2, and MSH6 was performed as previously described[53] to determine microsatellite stability status. Monoclonal antibodies used were as follows: MLH1 (Clone ES05, DAKO), PMS2 (Clone EP51, DAKO), MSH2 (Clone FE11, DAKO), and MSH6 (EP49, DAKO). IHC staining was evaluated as negative when all the tumor cells showed loss of nuclear staining; tumors with one negative staining of these four markers were considered as low level of microsatellite instable (MSI-L); tumors with two or more negative staining of these four markers were considered as high level of microsatellite instable (MSI-H); tumors with all 4 positive staining were defined as MSS.

IHC staining of CD8 (Clone SP16, ZSGB-BIO) were annotated within intratumoral areas; CD8+ density was quantified using Aperio Scanscope (Aperio Technologies Vista, CA, USA) by the method of rare event tissue test. The total number of CD8+ cells in each tumor area was counted based on six random captured visual fields (400×400 m$^2$), and the density of CD8+ T cells was defined as the total cell number per square millimeter.

IHC staining of IDO (Clone D5J4E, CST) was scored numerically on the intensity of IDO cytoplasmic staining (−, +, ++, +++) and percent tumor cells staining positive (−: 0%, +: 1–33%, ++: 34–66%, +++: ≥66%). All scoring was performed by two independent expert gastrointestinal pathologists (Z.L. and Yumei Lai), who were blind to clinical outcomes or proteomic cluster results.

**EBV infection status**. Chromogenic in situ hybridization with EBV-encoded small RNA (EBER) was performed to detect EBV infection using fluorescein-labeled oligonucleotide probes (INFORMEBER Probe; Ventana). Specimen in which EBER nuclear expression was observed in >20% of the tumor cells were considered EBER positive.

**Protein extraction and trypsin digestion**. Samples were minced and lysed in lysis buffer (8 M Urea, 100 mM Tris Hydrochloride, pH 8.0) containing protease and phosphatase Inhibitors (Thermo Scientific) followed by 1 min of sonication (3 s on and 3 s off, amplitude 25%). The lysate was centrifuged at 14,000×g for 10 min and the supernatant was collected as whole tissue extract. Protein concentration was determined by Bradford protein assay. Extracts from each sample (100 μg protein) was reduced with 10 mM dithiothreitol at 56 °C for 30 min and alkylated with 10 mM iodoacetamide at room temperature in the dark for additional 30 min. Samples were then digested using the FASP method[54] with trypsin; tryptic peptides were separated in a home-made reverse-phase C18 column in a pipet tip. Peptides were eluted and separated into nine fractions using a stepwise gradient of increasing acetonitrile (6%, 9%, 12%, 15%, 18%, 21%, 25%, 30%, and 35%) at pH 10. The nine fractions were combined to six fractions, dried in a vacuum concentrator (Thermo Scientific), and then analyzed by liquid chromatography tandem mass spectrometry (LC-MS/MS).

**LC-MS/MS analysis**. Samples were analyzed on Orbitrap Fusion, Orbitrap Fusion Lumos, and Q Exactive Plus mass spectrometers (Thermo Fisher Scientific, Rockford, IL, USA) coupled with an Easy-nLC 1000 nanoflow LC system (Thermo Fisher Scientific), or a Q Exactive HF mass spectrometer (Thermo Fisher Scientific, Rockford, IL, USA) connected to an UltiMate 3000 RSLCnano System (Thermo Fisher Scientific). Dried peptide samples were re-dissolved in Solvent A (0.1% formic acid in water) and loaded to a trap column (100 μm × 2 cm, home-made; particle size, 3 μm; pore size, 120 Å; SunChrom, USA) with a max pressure of 280 bar using Solvent A, then separated on a home-made 150 μm × 12 cm silica microcolumn (particle size, 1.9 μm; pore size, 120 Å; SunChrom, USA) with a gradient of 5–35% mobile phase B (acetonitrile and 0.1% formic acid) at a flow rate of 600 nl/min for 75 min. The MS analysis for QE HF and QE Plus were performed with one full scan (300−1400 m/z, R = 60,000 at 200 m/z) at automatic gain control target of 3e6 ions, followed by up to 20 data-dependent MS/MS scans with higher-energy collision dissociation (target 2 × 10$^3$ ions, max injection time 40 ms, isolation window 1.6 m/z, normalized collision energy of 27%), detected in the Orbitrap (R = 15,000 at 200 m/z).

For detection with Fusion or Fusion Lumos mass spectrometry, a precursor scan was carried out in the Orbitrap by scanning m/z 300−1400 with a resolution of 120,000 at 200 m/z. The most intense ions selected under top-speed mode were isolated in Quadrupole with a 1.6 m/z window and fragmented by higher energy collisional dissociation (HCD) with normalized collision energy of 35%, then measured in the linear ion trap using the rapid ion trap scan rate. Automatic gain control targets were 5 × 10$^5$ ions with a max injection time of 50 ms for full scans and 5 × 10$^3$ with 35 ms for MS/MS scans. Dynamic exclusion time was set as 18 s. Data were acquired using the Xcalibur software (Thermo Scientific).

**Peptide identification and protein quantification**. Raw files were searched against the human National Center for Biotechnology Information (NCBI) Refseq protein database (updated on 04-07-2013, 32,015 entries) by Mascot 2.3 (Matrix Science Inc) implemented in Proteome Discoverer 1.4 (Thermo Scientific). The mass tolerances were 20 ppm for precursor and 50 mmu for product ions from Q Exactive Plus and Q-Exactive HF, and 20 ppm for precursor and 0.5 Da for product ions for Fusion and Q-Exactive HF, respectively. Up to two missed cleavages were allowed. The search engine set cysteine carbamidomethylation as a fixed modification and N-acetylation, oxidation of methionine as variable modifications. Precursor ion score charges were limited to +2, +3, and +4. The data were also searched against a decoy database so that protein identifications were accepted at a false discovery rate of 1%. Label-free protein quantifications were calculated using a label-free, intensity-based absolute quantification (iBAQ) approach[18].

Proteins with at least 2 unique peptides with 1% FDR at the peptide level and Mascot ion score greater than 20 were selected for further analysis. Among all 11,340 proteins of this proteomics dataset, 9186 proteins met this requirement. The FOT was used to represent the normalized abundance of a particular protein across samples. FOT was defined as a protein's iBAQ divided by the total iBAQ of all identified proteins within one sample. The FOT was multiplied by 10$^5$ for the ease of presentation (Supplementary Data 2). We analyzed the 1008 (168 × 6) raw files together for uniformed quality control and protein identification with 1% global protein FDR. The file used for protein inference and protein FDR calculation was derived from Mascot search results, and the peptide spectrum match (PSM) was filtered via Percolator and customized parameters, and then the proteins were assembled. The protein FDR was calculated depending on the ratio of NPD (the number of assembled proteins from decoy database searches) and NPT (the number of assembled proteins from target database searches). In this dataset, the FDR of PSMs was 0.08% and FDR of peptide was 0.09%.

**MS Platform QC and DGC proteome QA**. For quality control (QC) of the MS performance, tryptic digestions of the 293T cell lysate were measured as QC standard every 2 days. The QC standard was made and run using the same method and conditions and same software and parameters for GC. Pairwise Spearman's correlation coefficient was calculated for all QC runs and the results are shown in Supplementary Fig. 2a. The average correlation coefficient among standards was 0.86 with the maximum and minimum of 0.99 and 0.76, respectively. The log$_{10}$ transformed FOTs for each GC sample (Supplementary Fig. 2b) were plotted to show consistency of data quality.

**Proteome data filtering and missing data imputation**. The following filter criteria were applied for each statistical analysis shown in Fig. 1a. (1) Dataset 1 (D1) included all 11,340 identified GPs on 1% of global FDR. (2) For dataset 2 (D2), proteins were required to have at least 2 unique peptides with 1% FDR at the peptide level and Mascot ion score greater than 20. Subcellular localization, molecular function type, and target drug(s) were annotated using Ingenuity® Pathway Analysis (IPA®, QIAGEN). (3) For dataset 3 (D3), we excluded keratins and proteins whose maximum FOT in all 168 experiments were less than 10$^{-5}$ in FOT, which was chosen as the minimum value according to Supplementary Fig. 3a and 3b. (4) For dataset 4 (D4), proteins were required to be identified in at least one-sixth (28) of all samples (either tumor or nearby tissue). (5) For dataset 5 (D5), the FOTs of all proteins whose FOT values were less than 10$^{-5}$ were replaced with 10$^{-5}$ to adjust extremely small values, and calculated the log 10 of tumor-nearby ratios. (6) For dataset (D6), proteins were required to have T/N ratio larger than 3 or less than 1/3, in at least 1/10 of patients (8 patients).

**Proteome data analysis**. PCA was performed to visualize separation of tumors and nearby tissues (Jolliffe 2002). SAM[55] (samr R package) was performed to find differentially expressed proteins between tumors and paired nearby tissues of all 84 patients and within each clusters (Supplementary Data 3). Other than the annotations mentioned above, we added target drug results from The Drug Gene Interaction Database (DGIdb 2.0)[56], stomach-related specific expressed type, and tissue from tissue-based human proteome map[20]. Data type was set as two class paired, not centered array data for SAM; delta value was set respectively to meet FDR <0.01. The differentially expressed genes defined here must meet the following criteria: (1) q-value less than 0.01, and (2) differentially expressed percentage larger than 50%, which was calculated using the following formula:

$$\text{Differentially expressed percentage} = \frac{|N_{\text{tumor up}} - N_{\text{tumor down}}|}{N_{\text{total detected}}}$$

where $N_{tumor\ up}$ means number of patients with T/N ratio larger than 3, $N_{tumor\ down}$ means number of patients with T/N ratio less than 1/3.

Fisher's exact test was used to find enriched gene sets/pathways (including 186 Kyoto Encyclopedia of Genes and Genomes (KEGG) gene sets[57], 217 Biocarta gene sets, 674 Reactome gene sets[58], 196 PID gene sets[59] and 50 cancer hallmark gene sets from MSigDB V5.1[60]) by differentially expressed genes (Supplementary Data 4). The 9186 genes detected in D2 were used as the background.

Wilcoxon rank-sum test was used to identify proteins with significantly different expression between mutant samples and wild-type samples and was also used to compare expression of stomach-specific and not stomach-specific proteins.

For mutation altered proteome, up-regulated proteins in mutated tumor tissues are: (1) differential expressed in mutated tumors and wild-type tumors ($P < 0.05$, Wilcoxon rank-sum test); (2) mean value of mutated tumors/wild-type tumors >1.6. Down-regulated proteins are: (1) differential expressed in mutated tumors and wild-type tumors ($P < 0.05$, Wilcoxon rank-sum test); (2) mean value of mutated tumors/wild-type tumors <0.33.

Jaccard index was used to calculate similarity among altered proteomes correlated to high-frequent mutants. Jaccard index between mutation altered proteome $i$ and proteome $j$ was defined as $\frac{altered\ proteomes\ i \cap altered\ proteomes\ j}{altered\ proteomes\ i \cup altered\ proteomes\ j}$, where $\cap$ denotes intersection between altered proteomes $i$ and $j$, and $\cup$ denotes union between altered proteomes $i$ and $j$.

**Proteome molecular subtyping of DGC**. Consensus clustering was performed using the R package ConsensusClusterPlus[61]. Samples were clustered using Euclidean distance as the distance measure. A total of 2538 proteins in D6 were used for $k$-means clustering with up to 6 clusters. The consensus matrices for $k = 2$, 3, 4 clusters are shown in Supplementary Fig. 4. The consensus matrix of $k = 3$ showed clear separation among clusters; the average silhouette width for $k = 2$, 3 (0.08) was higher than $k = 4$ (0.06), indicating stronger cluster separation. The cumulative distribution function (CDF) of the consensus matrix for each $k$-value was also measured (Supplementary Fig. 4b and 4c). Clustering by $k = 3$ had the lowest proportion of ambiguous clustering (PAC). The relative change in area under the CDF curve increased 30% from 2 clusters to 3 clusters, while others had no appreciable increase. Taken together, proteome clusters were defined using k-means consensus clustering with k = 3.

**Survival analysis**. All survival analysis used Kaplan−Meier method and the difference was tested using the log-rank test. Coefficient value, which equals to ln (HR), was calculated from Cox proportional hazards regression analysis. $P$-values less than 0.05 were considered as significantly different. OS was used as primary endpoint. Clinical variables analyzed with $P$-value less than 0.05 using single variant analysis were chosen to enter Cox regression multivariate analysis. The SPSS 22.0 software (IBM Corp.) and the R package "survival" was used for survival statistical tests.

**Nomination of prognosis-related druggable candidates for DGC**. We used two steps to nominate drug target candidates for DGC. First, we picked proteins that were overexpressed in tumors; second, we selected overexpressed proteins that were associated with worse OS to screen prognosis unfavorable therapeutic targets. A total of 159 genomic data discovered druggable cancer drivers/candidates were evaluated[26–28]. The drugs are comprised of (1) the Food and Drug Administration (FDA)-approved drugs, including direct targeting, indirect targeting, gene therapy, strong off-target and mild off-target; (2) drug in clinical trials, including direct targeting, indirect targeting and gene therapy; (3) pre-clinical ligand; (4) potentially druggable; and (5) potentially biopharmable.

**Targeted exome sequencing**. A capture panel was developed, which covered coding exons and flanking splicing junctions for 274 gastric cancer driver genes (Supplementary Data 2). This gene list collected all the significantly mutated, amplified, deleted genes in gastric cancer, especially diffuse-type gastric cancer, from three model studies[8, 10, 11] as well as all 138 cancer driver genes[19]. For each pair of tumor and paired nearby samples, genomic DNA was extracted either manually or automatically using the Gentra Puregene (Qiagen). DNA concentration was measured by a NanoDrop 1000 spectrophotometer (Thermo Scientific, Wilmington, DE). Briefly, 1 μg of genomic DNA from each sample was mechanically sheared, end repaired, and ligated to molecularly bar-coded adaptors to generate sequencing libraries following the manufacturer's standard protocol (Illumina). Co-capture was performed on pooled DNA libraries in groups of up to 48 samples. Captured sample DNA was sequenced on an Illumina HiSeq 2000 according to the standard operating protocol.

Paired-end reads in Fastq format were aligned to the reference human genome (University of California, Santa Cruz (UCSC) Genome Browser[62], hg19) using Burrows−Wheeler Aligner (BWA)[63]. Aligned reads were further processed following the GATK Best Practices of duplicate removal[64], indel realignment, and recalibration. Somatic single-nucleotide variations (SNVs) and small insertions and deletions (Indels) were detected by MuTect[65] and Pindel[66], respectively. In addition, variants were filtered against the ExAC[67] database using a cut-off of 0.1%. SNVs and Indels were annotated using SnpEff[68] based on UCSC known genes. Of the 274 exome targeted sequencing genes, 183 met the requirements of variant

allele frequency more than 0.05 in tumor tissues and less than 0.03 in paired normal tissues and were selected for further analysis. OncoPrint[69] was used to show 39 mutant genes with nonsynonymous mutation rate higher than 5% (Fig. 1d).

Mutation mapper[69, 70] was used to map location and frequency of mutations for *TP53*, *ATM*, and *ARID1A*, for these three genes showed significantly changed protein expression after mutation (Supplementary Fig. 5a), as well as other highly mutated genes in our cohort. Genes' mutation profile was used to generate pathways' mutation profile in each patient. A pathway was mutated when at least one gene in the pathway was mutated. Tested pathways included all gene sets from Canonical Pathways from MSigDB V5.1[60].

**Western blot**. For each sample, 20 μg of protein extracts from the previous step (See section Protein extraction and trypsin digestion) was separated on 10% sodium dodecyl sulfate−polyacrylamide gel electrophoresis. The proteins were transferred onto nitrocellulose membranes. After blocking with 5% milk (BD Science) solution in TBST (Tris buffered saline with Tween) for 1 h, the membranes were incubated with 5% milk containing appropriate primary antibodies overnight at 4 °C followed by 2 h of incubation with horseradish peroxidase-conjugated secondary antibodies. Signals of target protein bands were detected using Chemiluminescent detection reagent. IDO1 antibody (CST #86630) and β-tubulin antibody (CW0098) were used in a 1:1000 dilution.

**Data availability**. The proteomics data is accessible in the PRIDE Archive under the accession number PXD008840. The targeted exome sequencing data is accessible in NCBI SRA under the accession number SRP131815.

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

## Acknowledgements

This work was supported by the National Program on Key Basic Research Project (973 Program, 2014CBA02000, 2014CBA02001, 2014CBA02002, 2014CBA02004, 2012CB910300), National High-tech R&D Program of China (863 program, 2015AA020108), National International Cooperation Grant (2014DFB30010, 2014DFA33160, 2012DFB30080), Beijing Natural Science Foundation (Z131100005213003), Chinese Ministry of Science and Technology (2016YFA0502500), National Key Research and Development Program of China (2017YFC1308900, 2017YFA0505102, and 2017YFC0908404), National Natural Science Foundation of China (31270822, 31770886, and 31700682), National Institute of Health (Illuminating Druggable Genome, U01MH105026), Shanghai Municipal Science and Technology Major Project (Grant No. 2017SHZDZX01), and a grant from the State Key Laboratory of Proteomics (SKLP-YA201401). We thank Dr. Xiaofeng Fu for insightful discussions and guidance in the initial phase of the project.

## Author contributions

J.Q., Lin S., B.Z., C.D., and F.H. directed and designed research; S.G., M.L, G.G., W.H., and T.G. performed research; X.X., S.G., Q.Z., J.F., J.Y., Lei S., C.Z., S.Y.J., J.M.C., A.J., and W.L. performed analyses of mass spectrometry data and quality control samples and adapted algorithms and software for data analysis; S.G., J.W., J.J., L.Z., Z.L., Yumei Lai, Yanyan L., Yilin L., J.G., and Ying H. coordinated acquisition, distribution, and quality evaluation of tumor and nearby tissue samples; R.C., Yumei Li., and Z.G performed DNA sequencing and sequencing data analysis; L.C., J.X., Yin H., Y.Q., T.S., and W.Z. contributed new reagent/analytic tools; J.Q., Y.W., C.D., X.X., and S.G. wrote the manuscript.

**Additional information**

**Competing interests:** The authors declare no competing financial interests.

