## [Peer Review File · Nature Communications]

Reviewers' comments:

Reviewer #1 (Remarks to the Author):

Manuscript is largely descriptive and many of conclusions are not well supported by presented data. Molecular subtypes are only moderately associated with clinical outcomes. Furthermore, the association is questionable. Overall study lacks novel findings. I do not recommend publication without thorough re-analysis of the data and additional data from new experiments and re-writing.

Authors generated interesting proteomic data from diffuse gastric cancer to address molecular and clinical heterogeneity and their association. In addition, authors analyzed the data to uncover potential druggable therapeutic targets.

This is a potentially interesting study. However, study suffers from premature analysis of the data and many conclusions are not well supported by the data.

Major concerns:

1. Authors removed 27 patients with lower stage (stage I and II) from KM plots and survival analysis in Figure 3 without proper justification. What is rationale for removing these patients? They stated that number of event was low in patients with stage I and II. Is it what ones can expected from patients with lower stage? However, in Figure 5, all of patients were included in analysis. KM plots according to stage should be included in supplementary data. Stage information is missing in excel file for clinical data.
2. Throughout manuscript, authors stated that PX1 and PX2 is sensitive to chemotherapy and PX3 is resistant to chemotherapy. However, there is no data supporting such claims in manuscript. Clinical data in Figure 3D are not sufficient enough to support such statement since authors failed to demonstrated that patients in PX1 and PX2 subtype are indeed sensitive to chemotherapy. Furthermore, subtype-specific chemosensitivity (or chemoresistance) should be validated by interaction test. All analyses related to chemosensitivity should be removed from manuscript unless authors provide solid evidence.
3. In figure 3D, prognosis of highly immunogenic PX3 subtype is poorest among three subtypes. This is not in good agreement with previous observation from many clinical studies. In general, cancer immunity is well associated with good prognosis as it indicates active immune activity against cancer cells (Zhang L, Coukos G, et al. *N Engl J Med.* 2003;348(3):203-213. Galon J, Pagès F, et al. *Science.* 2006;313(5795):1960-1964.). Authors should provide in-depth discussion on this matter or reanalyze the data with inclusion of all patients.
4. In p11, authors stated that mutation status is not well correlated with protein expression. Since some proteins showed correlation, authors should carry out further analysis to address the difference. For examples, any correlation with mutation types? Vast majority of mutations in ARID1A is nonsense mutations. It would lead to lower translation of protein.
5. While it is interesting to see three molecular subtypes and their potential association of clinical outcomes. These findings should be further validated in independent cohort. TCGA study also generated limited proteomic data with RPPA technology. Authors should consider use of TCGA data as validation cohort.
6. What is rationale for using different cut-off for patient stratification in Figure 5c. For CD300A, cut-off doesn't appear to be median. Patients were split to 23 vs. 59.
7. Analysis of proteomic data for genomic targets (p15) is not informative. Since authors already showed that mutation status is not correlated with protein expression in p11, it is highly expected to see low correlation in protein expression. However, it does not necessarily indicate that they are not good therapeutic targets. Thus, this cannot be used as supporting evidence for that proteomic data is superior to genomic data for finding therapeutic targets. They are complimentary to each

other. Authors may consider to remove this section from manuscript.

8. Overall, study also lacks functional validation of findings in cell culture models.

9. Proteomic data should be validated with western blots using same tissues. At least test few proteins with clinical significance. IHC is not really quantitative.

Minor

1. In page 22 line 417, ATGC should be TCGA

Reviewer #2 (Remarks to the Author):

This study has completed a detailed proteomic analysis of 84 DGCs and adjacent normal tissue and used that information to identify 3 classes of tumour with distinct characteristics defined by the presence/absence of an EMT or immune response. The 'immune' group (PX3) was the largest and shown to be resistant to chemotherapy, providing a rationale for improved targeting of DGC treatment. The proteomic analysis showed the clear loss of differentiation in this tumour subtype; although not unexpected, this detailed proteome level data is a valuable representation of this phenotype. The samples also underwent targeted DNA sequencing to identify common mutations and correlate these mutations with the protein expression pattern. The study, which is the most thorough proteomic analysis of DGC to date, provides a useful resource for further research into DGC.

Questions/additional comments

It is unfortunate that RNAseq (or even Affymetrix arrays) was not carried out on the samples since a comparison of the correlation between protein and RNA levels would have been of considerable interest, particularly if the influence of DNA mutations was overlaid on the data.

It was not clear if tumour heterogeneity accounted for any of the discrepancy between driver gene mutations and the detection of their encoded proteins in the proteome. If a higher threshold for the variant allele is used, does the correlation improve? Eg if an oncogenic mutation is only present in 10% of the tissue, will the proteomic analysis detect it? Clarification of this is particularly important given the strength of the statements made in lines 217-222. It might also help with the interpretation of the Jaccard index data (lines 264-266) and the correlations between mutations and survival (lines 286-187).

CDH1 was used as an example to explore the association between mutations and alterations in the proteome of cancer pathways. Because the consequences of many CDH1 missense mutations aren't well understood, it would have been interesting to also examine the effects of truncating mutations alone.

Only 11/34 tumours from the PX3 group had elevated CD8+ T cells, and there was no difference between the TIL percentages of the three groups. Is it therefore justifiable to label the PX3 class as the 'immune response' class?

The identification of proteomic targets included the requirement that a protein needed to be differentially over-expressed in 50% of tumours. This is therefore asking for targets/drugs that are potentially useful as a generic DGC treatment, and pays no heed to the three subtypes identified by the authors. If possible, it would be useful to repeat the target identification for each of the three subtypes separately.

Is it possible to comment on the overlap between the three subtypes identified here and the 4 class classifications made by ATGC/ACRG?

Reviewer #3 (Remarks to the Author):

In the manuscript by Ge et al., the authors presented proteomic dataset from 84 diffuse-type gastric cancer-normal pairs. Data analysis showed diffuse-type gastric cancer could be subtyped into three major classes based on proteomic data. The data was further mined to identify proteins that may be targeted for therapy. This dataset provides rich information for study biology of diffuse-type gastric cancer. The study, including study design, workflow, data analysis, and dataset, will be great interesting to cancer biologists and clinicians. With further considerations of detailed description/clarification list below, the manuscript should be acceptable.

1. Analysis of diffuse-type gastric cancer is generally challenging due to the mix of cancer and normal cells and low cellularity. How did this study control the consistency of cellularity .
2. Author list should start with performance site #1 instead of #2.
3. In the author list, who else was CHHP referring to? The CNHPP names should be included in the end of the manuscript or CNHPP should be deleted from author list.
4. In the introduction, page 4, line 7, the 4 subtypes are GC subtypes or DGC subtypes?
5. For spectrum assignment, peptide and protein identifications, the FDR descriptions were not clear, for example, in result session (page 6), 1% FDR was referring to global protein FDR, what is the FDR in peptide and spectrum levels?
6. In result session (page 7), the authors described that one of the filters was to pick proteins that were detected in at least 1/6 of the samples (28 cases), what was the rationale for this filter and other filters that were applied to the data analysis.
7. In result session (page 7), CVs and FOT of the proteins were analyzed through the samples. These analyses should also performed on the data for the quality control runs in the whole data-collecting period. Is the FOT value consistent throughout the whole data-collecting period? Will different FOT values be used based on the instrument performances or mass spectrometers? This should be assessed by the quality control data.
8. In the result session (page 7), 3 folds was selected as cut-off for differentially expressed proteins. Why 3 folds was selected to filter the quantitative data.
9. In result session (page 10), the authors stated that "it was remarkable that cancer subtyping and association with clinical outcome could be achieved solely based on the altered cancer proteome." However, the stability of these subtypes is not confirmed. The subtypes might depend on how the proteomic features were selected.
10. In result session (page 11), " Among the 183 mutated genes, only 16 gene products demonstrated altered expressions compared with the wild type (WT), suggesting that mutations do not always strongly impact protein expression." This statement needs further support or clarification, for example, the lack of protein changes of other mutated genes could due to the small number of samples with the mutations.
11. In method session (page 26), 428 were treated with neoadjuvant chemotherapy or chemoradiation therapy before operation. The treatment for the samples used in this study should be summarized.
12. In method session, were both normal and cancer pieces genomically analyzed?
13. In method session (page 30), Orbitrap Fusion mass spectrometer and Q Exactive HF mass spectrometer were used to collect MS/MS data, how would different mass spectrometers affect the intensity cut off and spectrum requirement for quantitation?
14. In method session (page 32), what would be the FDR for peptide and spectrum or protein levels? Were FDR applied to the entire assembled data set or to individual run or sample?
15. The availability of the datasets to the public, and access information of the data should be described in details.

Reviewers' comments:

Reviewer #1 (Remarks to the Author):

Manuscript is largely descriptive and many of conclusions are not well supported by presented data. Molecular subtypes are only moderately associated with clinical outcomes. Furthermore, the association is questionable. Overall study lacks novel findings. I do not recommend publication without thorough re-analysis of the data and additional data from new experiments and re-writing.

Authors generated interesting proteomic data from diffuse gastric cancer to address molecular and clinical heterogeneity and their association. In addition, authors analyzed the data to uncover potential druggable therapeutic targets.

This is a potentially interesting study. However, study suffers from premature analysis of the data and many conclusions are not well supported by the data.

We thank the reviewer for the comment and disagree with this reviewer on the general assessment by this reviewer on the significance of our study. The following is our answer to the specific concerns raised by this reviewer.

Major concerns:

1. Authors removed 27 patients with lower stage (stage I and II) from KM plots and survival analysis in Figure 3 without proper justification. What is rationale for removing these patients? They stated that number of event was low in patients with stage I and II. Is it what ones can expected from patients with lower stage? However, in Figure 5, all of patients were included in analysis. KM plots according to stage should be included in supplementary data. Stage information is missing in excel file for clinical data.

RESPONSE: 1) We originally did KM plot of all patients, as shown below (Figure RL1). However, due to the short follow-up time, only 1 of 25 patients at earlier stage (stage I and II) died, whereas 24 of them were censored at the end of follow-up. Since 22 of the 57 patients with advanced stage (stage III and IV) died, we included only advanced stage patients in the KM plots, trying to show the correlation between overall survival and the proteomics subtypes based on less than three years' follow-up (Fig. 3D). We agreed with the reviewer and added KM plot of all patients in supplementary Fig. 4E.

A longer follow-up time would provide stronger support for the relationship between subtyping and overall survival of patients at all stages.

Figure RL1/Supplementary Figure 4E. Molecular subtypes and overall survival of all patients.

(Kaplan-Meier analysis, P value from Log-rank test)

2) As for the question “in Figure 5, all of patients were included in analysis”, we included all patients in nominating druggable protein targets because we wanted to find the widely applicable drug targets to treat all DGC patients regardless of proteomic subtyping nor TNM staging.

3) Regarding “Stage information is missing in excel file”, the stage information could be found in the Supplementary Data 1 (excel file), in column U (TNM stage) and column Y (pStage). The classification was done according to AJCC (7th edition).

2. Throughout manuscript, authors stated that PX1 and PX2 is sensitive to chemotherapy and PX3 is resistant to chemotherapy. However, there is no data supporting such claims in manuscript. Clinical data in Figure 3D are not sufficient enough to support such statement since authors failed to demonstrated that patients in PX1 and PX2 subtype are indeed sensitive to chemotherapy. Furthermore, subtype-specific chemosensitivity (or chemoresistance) should be validated by interaction test. All analyses related to chemosensitivity should be removed from manuscript unless authors provide solid evidence.

RESPONSE: We acknowledge that the number of patients who did not receive adjuvant chemotherapy in PX1 and PX2 is too small to make a statistically sound conclusion on response to chemotherapy. For the PX3 subgroup (stage III/IV, KPS \geq 90), 16 patients were treated with adjuvant chemotherapy and 8 were not treated with adjuvant chemotherapy, which allowed us to perform a statistical evaluation on the benefit of the chemotherapy. The statistical analysis indicates that there is no significant difference between chemotherapy group and the non-chemotherapy group in PX3.

We acknowledge that any conclusions about “chemo-sensitivity” needs to be validated experimentally (ideally in a perspective trial setting), and therefore stated

that PX3 patients may not benefit from classical chemotherapy and hence were in need of alternative treatments, such as immunotherapies.

In the revised version, we have tuned down our statement and removed “Our analyses indicated that PX1 plus PX2 population may benefit from chemotherapy, and PX3 may be the prime target population for immunotherapy” and all related statements in the manuscript.

3. In figure 3D, prognosis of highly immunogenic PX3 subtype is poorest among three subtypes. This is not in good agreement with previous observation from many clinical studies. In general, cancer immunity is well associated with good prognosis as it indicates active immune activity against cancer cells (Zhang L, Coukos G, et al. N Engl J Med. 2003;348(3):203-213. Galon J, Pagès F, et al. Science. 2006;313(5795):1960-1964.). Authors should provide in-depth discussion on this matter or reanalyze the data with inclusion of all patients.

RESPONSE: 1) It is true that tumor-infiltrating immune cells, for example, CD8+/CD3 T cells is associated with good prognosis in ovarian, colon, breast cancer and other solid tumor types (Zhang L, Coukos G, et al. N Engl J Med. 2003; 348(3): 203-213. Galon J, Pagès F, et al. Science. 2006; 313(5795):1960-1964.)^{1, 2}. However, in gastric cancer it is different. In a reference (Thompson ED, Zahurak M, et al. Gut. 2017; 66(5):794-801)³, increased CD8 infiltration is correlated with impaired PFS and OS in gastric cancer and gastro-esophageal junction (EGJ) cancers, and patients with higher CD8+ T cell densities also have higher PD-L1 expression. The reason for such difference is not clear, but it may indicate an adaptive immune resistance mechanism. For the 84 patients in our study, we analyzed the correlation of CD8+ T cells and overall survival. As shown below (Figure RL2), an increase in number of CD8+T cells is not significantly correlated with an increase of overall survival in this cohort.

Figure RL2. CD8+ T cell number is not associated with overall survival. (A) Definition of Low, medium and high level of CD8+ T cell number. (B) CD8+ T cells number is not significantly correlated with patients' overall survival (Kaplan-Meier analysis, P value from Log-rank test)

2) Our proteomic result showed that the PX3 subgroup with the worst prognosis is characterized with an enrichment of immune related proteins, such as IDO1, ARG1, SIRPA and TMEM173. We termed it immunological process enrichment subtype for this reason. This is not to be confused with cancer immunity that is defined by measurements from IHC of tumor-infiltrating immune cells, such as CD8+/CD3 T cells. The PX3 proteomics subtype may help us better understand tumor immune evasion and immunotherapy for gastric cancer.

To avoid confusion, we changed PX3 subtype to “immunological process enrichment subtype”.

4. In p11, authors stated that mutation status is not well correlated with protein expression. Since some proteins showed correlation, authors should carry out

further analysis to address the difference. For examples, any correlation with mutation types? Vast majority of mutations in ARID1A is nonsense mutations. It would lead to lower translation of protein.

RESPONSE:

- 1) We investigated all types of nonsynonymous mutations (termed as “mutated”) and used Wilcoxon rank-sum test to determine whether a significant difference of protein expressions (T-N ratio) exists between the mutated and not-mutated samples. We found that 9 genes showed significant difference. Thus, while *MED12* and *NF1* showed higher protein expression in mutated samples ($FC > 2$, $P < 0.05$), *ATM*, *BAX*, *ARID1A*, *SOX9*, *PLEKHA6*, *FLII* and *NCOR1* showed lower protein expression in mutated samples (Figure RL3A). For *ARID1A*, all mutations are protein truncating mutations, which include 5 nonsense mutations and 6 frame-shift InDels, and *ARID1A* mutated samples had lower *ARID1A* protein expression ($P < 0.001$).
- 2) If we focus only on protein-truncating mutations, which include nonsense mutations, frame-shift deletions, frame-shift insertions and exon-loss mutations, we found that 4 genes (*RBMX*, *ARID1A*, *SOX9* and *TMPO*) had significantly lower protein expression (Figure RL3B).

We have added the analysis of correlation between protein truncating mutations and protein expressions in the revised version (Figure RL3A/Supplementary Fig. 5C, Figure RL3B/Supplementary Fig. 5D).

Figure RL3/Supplementary Figure 5C, D. Association between mutations and their protein expressions. (A). A volcano plot to illustrate significantly altered protein expressions for genes with nonsynonymous mutations compared with the wild type. Up-regulated and down-regulated genes are indicated in red and blue, respectively (fold change > 2 and P value < 0.05). P values were calculated using Wilcoxon rank sum test. (B). A volcano plot to illustrate significantly altered protein expressions

comparing genes with/without protein-truncating mutations. Up-regulated and down-regulated genes are indicated in red and blue, respectively (fold change > 2 and P value < 0.05). P values were calculated using Wilcoxon rank-sum test.

5. While it is interesting to see three molecular subtypes and their potential association of clinical outcomes. These findings should be further validated in independent cohort. TCGA study also generated limited proteomic data with RPPA technology. Authors should consider use of TCGA data as validation cohort.

RESPONSE: The TCGA-GC RPPA dataset measured 138 proteins in total, and of those, 105 were quantified in our study. We found that only 25 of the 105 proteins are differentially expressed in the three subgroups of our dataset (T/N ratio among PX1~3, ANOVA $P < 0.05$). Since TCGA study contained tumor tissues only, we could not use T/N ratio for the analysis; we then searched for differentially expressed proteins among the three proteomic subtypes (PX1~3) using only tumor data, and found that 5 proteins were expressed differently among the three subtypes.

We tried to test whether we could use these 5 proteins to achieve subtyping using a machine learning method. We randomly divided the 84 samples into 62 training samples (2/3 of the samples) and 22 test samples (1/3), and used the 5 proteins to train a SVM (Supporter Vector Machine) model implementing linear kernel using training samples. The accuracy of predicting the test samples with the SVM model is only 53.4%.

Failed to find a good classifier with the 5 proteins, we then used all 105 proteins in an attempt to generate a better classifier. We used the LASSO algorithm for this purpose. Figure RL4A showed that the average error rate decreased when more proteins were included in the classifier from the 62 training samples; unfortunately the error rate was around 40%, which was too high. Figure RL4B showed the average error rate of around 50% for predicting the 22 test samples.

We thus concluded that it was not feasible to use the TCGA data as a validation cohort.

Figure RL4. Average error rate with top 2~10 proteins in training data (A) and test data (B) in 100-time sampling procedure. Proteins were ranked according to appearing frequency in 100-time sampling procedure by LASSO.

6. What is rationale for using different cut-off for patient stratification in Figure 5c. For CD300A, cut-off doesn't appear to be median. Patients were split to 23 vs. 59.

RESPONSE: In Figure 5C, we used a uniform cut-off for each protein, which is the median value of $\log_{10}(T/N)$ ratio among all patients. We replaced all missing data with the lowest expression value 10^{-7} . For CD300A, 57 patients had no detectable values in neither tumor nor normal tissues, 23 patients had higher expression in tumor, and 2 patients had lower expression in tumors. Thus, the median $\log_{10}(T/N)$ ratio value of CD300A was zero, and patients were split into 23 vs. 59.

7. Analysis of proteomic data for genomic targets (p15) is not informative. Since authors already showed that mutation status is not correlated with protein expression in p11, it is highly expected to see low correlation in protein expression. However, it does not necessarily indicate that they are not good therapeutic targets. Thus, this cannot be used as supporting evidence for that proteomic data is superior to genomic data for finding therapeutic targets. They are complimentary to each other. Authors may consider to remove this section from manuscript.

RESPONSE: It is true that analysis of proteomic data for genomic targets does not help reconcile the difference between these 2 datasets, and indeed, genomic analysis has provided critical clues leading to the finding of therapeutic targets. However, we believe that using high protein expression level in combination with poor overall

survival may provide an approach for more efficient therapeutic targeting. We now named these proteins as “prognosis unfavorable targets”. A similar definition has been also used in a recently published paper⁴. We also have rewritten this section in the main text.

8. Overall, study also lacks functional validation of findings in cell culture models.

RESPONSE: Like several published proteomic studies (Mertins, P. et al. Nature 534, 55-62; Zhang, H. et al. Cell 166, 755-765; Wilhelm, M. et al. Nature 509, 582-587; Zhang, B. et al. Nature 513,382-387)⁵⁻⁸, the main goal of this paper is to dissect the proteomic landscape of diffuse-type gastric cancer, expanding the knowledge of DGC from a proteomic perspective. Most of the CPTAC publications did not have validation data. In this paper, IHC and western blotting of IDO-1 were performed on clinical samples to validate the accuracy of mass spectrometric quantification as suggested by the reviewer. Functional and biochemical validations using cell culture and PDX mouse models will be our focus in the future.

9. Proteomic data should be validated with western blots using same tissues. At least test few proteins with clinical significance. IHC is not really quantitative.

RESPONSE: According to reviewer’s suggestion, we performed western blotting of IDO1 on the same samples which the IDO1 IHCs were done and obtained consistent results. Figure RL5 was added as Figure 6C in the revision.

Figure RL5/figure 6C: IHC and western blot analysis of IDO1 in selected patients' tumor samples.

Minor

- 1. In page 22 line 417, ATGC should be TCGA**

RESPONSE: Changed as suggested.

Reviewer #2 (Remarks to the Author):

This study has completed a detailed proteomic analysis of 84 DGCs and adjacent normal tissue and used that information to identify 3 classes of tumour with distinct characteristics defined by the presence/absence of an EMT or immune response. The 'immune' group (PX3) was the largest and shown to be resistant to chemotherapy, providing a rationale for improved targeting of DGC treatment. The proteomic analysis showed the clear loss of differentiation in this tumour subtype; although not unexpected, this detailed proteome level data is a valuable representation of this phenotype. The samples also underwent targeted DNA sequencing to identify common mutations and correlate these mutations with the protein expression pattern. The study, which is the most thorough proteomic analysis of DGC to date, provides a useful resource for further research into DGC.

Questions/additional comments:

1. It is unfortunate that RNAseq (or even Affymetrix arrays) was not carried out on the samples since a comparison of the correlation between protein and RNA levels would have been of considerable interest, particularly if the influence of DNA mutations was overlaid on the data.

RESPONSE: We agree that a comparison between protein and RNA level would be informative. We did not perform this analysis from two considerations: (1) we failed to obtain high quality RNA samples for RNA-seq. We did not store samples in reagents such as RNAlater and found that easy degradation of RNA is a serious problem particularly for gastric cancer samples, presumably caused by the high level of digestive enzymes; (2) Many proteo-genomic investigations that investigated the correlation between RNA-seq and protein profiling have shown that the overall correlation was not very good. We aimed to investigate whether protein profiling data alone was sufficient to subtype DGC and were pleased to find that the answer is YES. We hope our finding that protein profiling is sufficient to subtype DGC will encourage the field to focus more on proteins as they are the executioners of biological functions after all, whereas RNA provides indirect information in most of the situations.

We did try to compare our protein data with the ACRG RNA seq data, but was not successful because 1) the ACRG RNA-seq was performed on tumor samples only; 2) since ours and ACRG were two different cohorts, a direct comparison was difficult to make.

2. It was not clear if tumour heterogeneity accounted for any of the discrepancy between driver gene mutations and the detection of their encoded proteins in the proteome. If a higher threshold for the variant allele is used, does the correlation improve? Eg if an oncogenic mutation is only present in 10% of the tissue, will the proteomic analysis detect it? Clarification of this is particularly important given the strength of the statements made in lines 217-222. It might also help with the interpretation of the Jaccard index data (lines 264-266) and the correlations between mutations and survival (lines 286-287).

RESPONSE: We agree that tumor heterogeneity should be taken into consideration in the data analysis. If a mutation is present in only a small amount of tumor, the protein level change would not be significant. To be more cautious in data interpretation, we added discussion about tumor heterogeneity.

As for the issue of threshold, we increased the threshold of the variant allele (VAT) from 5% to 10, 15 and 20% and then recalculated the correlations. For some genes (*RBMX* and *TP53*), increasing of VAT showed expression difference between mutants and wild types, but for other genes (*FLI1*, *NF1* and *MED12*), increasing of VAT erased the correlation between mutation and expression, mainly because of smaller mutated sample size.

Among all 183 mutated genes, very limited number of genes showed positive correlation between mutation status and protein expression, even after increasing of VAT (Table RL1, Figure RL6). Thus, our conclusion remains that the correlation between mutations and protein expression was poor. As 5% is a commonly used variant allele threshold⁹ (for example, Cancer Genome Atlas Research Network. *Nature*. 2017; 541(7636):169-175), we still described the result under VAT of 5% in the main text and provided results under VAT of 10%-20% in the table below and the Rebuttal Letter Supplementary Data for this reviewer only.

Symbol	Variant allele threshold			
	0.05	0.1	0.15	0.2
ARID1A	down	down	down	down
NCOR1	down	down	down	down
ATM	down	down	down	down
PLEKHA6	down	down	down	-
SOX9	down	down	down	-
FLI1	down	-	-	-
BAX	down	down	down	-
RBMX	-	-	down	down
NF1	up	up	-	-
MED12	up	up	-	-
TP53	-	up	up	up

Table RL1. Differentially expressed proteins between mutated samples and wild types

with different mutation frequency threshold (“down” denoted mutated samples with lower protein expression, “up” denoted mutated samples with higher protein expression, “-“ denoted no significance).

Figure RL6. Volcano plots to illustrate significantly altered protein expressions for genes with nonsynonymous mutations compared with the wild types (Variant allele threshold for A~D is 0.05, 0.1, 0.15 and 0.2, respectively). Up-regulated and down-regulated genes are indicated in red and blue, respectively. *P* values were calculated using Wilcoxon rank sum test.

For Jaccard index data, increasing the VAT improved the correlation between altered proteomes of *APC* and *PIK3CA* (Figure RL7 A-H), also improved correlation between *SPTA1* and *PIK3CA*. For the same reason above that 5% is commonly used VAT, we still described the result under VAT of 5% in the main text (Figure RL7 A,B/Supplementary Fig. 5F,G and provided result under VAT of 10%-20% in the Rebuttal Letter Supplementary Data (Figure RL7C-H).

Figure RL7. Altered proteomes associated with nine high-frequency mutations compared with wild types. (A,C,E,G) Hierarchical clustering of altered proteomes associated with nine high-frequency mutations (VATs for A,C,E,G are 0.05, 0.1, 0.15 and 0.2) compared with wild types. Colors indicate ratio of median tumor expression in mutated samples to that in the wild type samples, numbers in brackets at the bottom indicate the number of altered proteins associated with each gene mutation. (B,D,F,H) Heat maps of altered proteome similarity for the nine high-frequency mutations (VATs for B, D, F, H are 0.05, 0.1, 0.15 and 0.2). Similarity between each pairs of mutations was estimated with the pair-wise Jaccard index.

Regarding the sentence “correlations between mutations and survival (lines 286-287)”, what we meant is that the genes listed by genomic data were not correlated with survival in this dataset, not that mutations were not related with survival.

3. CDH1 was used as an example to explore the association between mutations and alterations in the proteome of cancer pathways. Because the consequences of many CDH1 missense mutations aren’t well understood, it would have been interesting to also examine the effects of truncating mutations alone.

RESPONSE:

As suggested by the reviewer, we compared 2 different types of mutations with wild type: (1) all nonsynonymous mutations, and (2) protein truncating mutations (including Nonsense Mutation, Frame Shift Del, Frame Shift Ins and exon loss). Examining only the truncating mutations indeed improved the correlation between *CDH1* mutation and its protein expression (Wilcox test *P* value = 0.099) compared with using all nonsynonymous mutations (Wilcox test *P* value = 0.45) (Figure RL8, Fig.4D).

Since only 4 *CDH1* mutations were truncating mutations, restricting to truncating mutations improved only the correlation of *CDH1*'s expression, but not the correlations of other proteins in the pathway analysis likely because of the smaller number of mutated samples (Table RL2). Hence, we did not change Figure 4D.

We have added the discussion about different mutation types in the revised manuscript.

Figure RL8. *CDH1* protein expression difference between mutated samples and other samples. A) Difference between *CDH1* expressions in samples with all types of *CDH1* mutations and other samples; B) Difference between *CDH1* expressions in samples with *CDH1* protein truncating mutations and other samples

Gene	All types of CDH1 mutation		Protein-truncating mutation		CDH1	
	P value	Average M	Average W	P value	Average M	Average W
LRP10	0.009**	-1.840	-1.379	0.115	-2.000	-1.470
CYTH1	0.158	-0.674	-1.075	0.275	-1.422	-0.951
IGSF8	0.008**	-2.000	-1.553	0.312	-2.000	-1.649
AKT2	0.012*	-0.251	-0.704	0.176	-0.112	-0.613
WNT5A	0.001***	-0.934	-1.565	0.090	-0.797	-1.436
FOCAD	0.043*	-0.711	-0.989	0.040	-0.448	-0.943
EGFR	0.039*	-0.414	-0.754	0.924	-0.740	-0.664
CDH1	0.454	0.392	0.402	0.099	-0.747	0.457
ASPN	0.037*	1.526	1.141	0.924	1.316	1.235
EPHB3	0.006**	-1.910	-1.507	0.161	-2.000	-1.589

Table RL2. Altered expression of proteins in Figure 4D when examining all types of *CDH1* mutations or truncating mutations only (*P* value, Wilcoxon rank sum test).

Average M, average tumor FOT value of mutated ones ($\log_{10}\text{FOT}+5$); Average W, average tumor FOT value in samples with wild type *CDH1* ($\log_{10}\text{FOT}+5$). * denotes $P<0.05$, ** denotes $P<0.01$, *** denotes $P<0.001$.

4. Only 11/34 tumours from the PX3 group had elevated CD8+ T cells, and there was no difference between the TIL percentages of the three groups. Is it therefore justifiable to label the PX3 class as the ‘immune response’ class?

RESPONSE: Our proteomic result showed that the PX3 subgroup with the worst prognosis is characterized with enrichment of immune related proteins, such as IDO1, ARG1, SIRPA and TMEM173. We therefore termed this subtype as immune subtype for this reason. This is not to be confused with cancer immunity that is defined by the measurements from IHC of tumor-infiltrating immune cells, such as CD8+/CD3+ T cells. Many immune related proteins that we identified in PX3 actually were enriched in macrophages, NK cells and antigen presenting cells. To avoid confusion, we changed PX3 subtype to “immunological process enrichment subtype”, and added this explanation in the discussion. The PX3 proteomics subtype may help us better understand tumor immune evasion, immunotherapy for gastric cancer.

5. The identification of proteomic targets included the requirement that a protein needed to be differentially over-expressed in 50% of tumours. This is therefore asking for targets/drugs that are potentially useful as a generic DGC treatment, and pays no heed to the three subtypes identified by the authors. If possible, it would be useful to repeat the target identification for each of the three subtypes separately.

RESPONSE: This is a great suggestion. We performed target identification for each subgroup, and the results were presented in Supplementary Fig.6. These results were displayed in Supplementary Data 6 sheet “prognosis unfavorable targets”.

6. Is it possible to comment on the overlap between the three subtypes identified here and the 4 class classifications made by ATGC/ACRG?

RESPONSE: This is a great suggestion, however, it’s hard to compare because first, our cohort only included diffuse-type gastric cancer (DGC) patients, and TCGA and ACRG cohort included DGC, intestinal-type gastric cancer (IGC) and mixed-type gastric cancer (MGC) patients; and second, we didn’t have DNA copy number data nor mRNA expression data. Still, we tried to make comparison of our subtypes and TCGA/ACRG subtypes using our 84 DGC data as follows:

1) Comparing with TCGA:

TCGA contained 295 GC patients, including 69 DGC patients (23.4%), 196

intestinal-type gastric cancer (IGC) patients (66.4%), and 19 mixed-type gastric cancer (MGC) patients (6.4%). TCGA proposed a molecular classification of gastric cancer into 4 subtypes: EBV positive, MSI, genomically stable (GS, mainly DGC) and chromosomal instable (CIN). The proportion of these four subtypes in our DGC cohort and TCGA-DGC/GC cohort are shown in Table RL3.

As shown in Figure 6C, all MSI-H patients (N=4) were in PX3, and the 6 EBV-positive patients were in PX2 and PX3, respectively (2 in PX2 and 4 in PX3); we could not classify CIN subtype in our cohort, because we did not have chromosomal copy number related data.

We did a proteomics analysis for DGC only, which is mainly the genomically stable (GS) subtype according to the TCGA classifications. Yet, we showed that the GS subtype can be further divided into three subtypes (PX1, PX2 and PX3) at the protein level, demonstrating the complexity of DGC and the power and higher resolution of cancer molecular subtyping with proteomic data.

TCGA subtype	our data	TCGA DGC	TCGA GC
EBV positive	6 (7.1%)	5 (7.2%)	26 (8.8%)
MSI	4 (4.7%)	6 (8.7%)	64 (21.7%)
GS	74 (88.1%)	40 (58.0%)	58 (19.7%)
CIN		18 (26.1%)	147 (49.8%)
Total	84	69	295

Table RL3. Proportion of 4 TCGA subtypes in different populations.

2) Comparing with ACRG:

ACRG contained 295 GC patients, including 45% DGC (135/300), 48.7% IGC (146/300) and 5.7% MGC patients (17/300).

Using gene expression profiling, copy number profiling and target sequencing data, the ACRG established clinical relevant molecular subtypes, including MSI, MSS/EMT, MSS/TP53 active and MSS/TP53 inactive. It should be noticed that in ACRG, DGC accounts for 80.4% (37/46) MSS/EMT subtype. The percentage of MSS/EMT in our cohort is 46.4% (39/84), while in ACRG cohort the percentage is 27.4% (37/135) in DGC and 15.3% (46/300) in all GC samples. This may be because of:

1) EMT markers listed by RNA data may not be all suitable for proteomic results; 2) different populations.

Of the 84 patients in our study, 4 MSI patients were all in PX3. MSI/EMT samples were predicted by consensus clustering using FOT of tumor samples with 43 EMT-up proteins and 57 EMT-down proteins provided by ACRG study (Figure RL9D). MSS/TP53 active and MSS/TP53 inactive subtypes was defined using

CDKN1A and MDM2 expression, since we did not have CDKN1A nor MDM2 quantity data, we could not further classify MSS/TP53 active and MSS/TP53 inactive subtypes. The overlapping between our protein subtypes and ACRG subtypes is shown in Table RL4.

It remains to be seen how protein subtyping correlates with DNA or RNA subtyping. We will know the answer when we are done with proteomics analysis of DGC, IGC and MGC combined.

Figure RL9. Identification of MSS/EMT subtype according to ACRG. (A) Consensus matrices of the 79 MSS GC samples ($k=4$), performed with 43 EMT-up proteins and 57 EMT-down proteins; (B) Consensus CDF plot of consensus clustering; (C) Delta area for each number of clusters; (D) A heatmap of 43 EMT-up proteins and 57 EMT-down proteins of all 79 samples, ordered by clustering result on Figure RL9A (yellow denotes expression is higher in tumors, while blue denotes expression is lower in tumors). The EMT cluster with higher “EMT up” expression and lower “EMT down” expression was tagged.

	Total(n)	PX1(n)	PX2(n)	PX3(n)
MSI-H	4	0	0	5
MSS/EMT	39	5	18	16
MSS/TP53+ or MSS/TP53-	41	11	16	14

Table RL4. Overlap between proteomic subtypes and ACRG subtypes.

Figure RL10. The association of ACRG classification with overall survival (Kaplan-Meier analysis, P value from Log-rank test).

Reviewer #3 (Remarks to the Author):

In the manuscript by Ge et al., the authors presented proteomic dataset from 84 diffuse-type gastric cancer-normal pairs. Data analysis showed diffuse-type gastric cancer could be subtyped into three major classes based on proteomic data. The data was further mined to identify proteins that may be targeted for therapy. This dataset provides rich information for study biology of diffuse-type gastric cancer. The study, including study design, workflow, data analysis, and dataset, will be great interesting to cancer biologists and clinicians. With further considerations of detailed description/clarification list below, the manuscript should be acceptable.

1. Analysis of diffuse-type gastric cancer is generally challenging due to the mix of cancer and normal cells and low cellularity. How did this study control the consistency of cellularity.

RESPONSE: We curated all DGC tumor tissues to ensure that tumor cells are present in >50% of the cells by H&E stain. Each tumor tissue used for profiling was first cut into four pieces, one for DNA-sequencing, one for FFPE and H&E staining, one for protein extraction and the last piece saved for future validation. As mentioned in the method (Section “Biospecimen collection”, paragraph 2, line 9-12), we eliminated 56 of the 146 pairs of DGC samples due to their low tumor purity (< 50%).

2. Author list should start with performance site #1 instead of #2.

RESPONSE: The first author, Sai Ge, who contributed most to this work, is a member of site #2 (Key Laboratory of Carcinogenesis and Translational Research, Peking University Cancer Hospital & Institute). She performed most experiments at site #1 (Beijing Proteomic Research Center (BPRC)) under the supervision of Dr. Jun Qin, who is affiliated with BPRC. Beijing Cancer Hospital has provided all patient samples and related clinical information. And Sai Ge was also under the direct supervision of Dr. Lin Shen, who is affiliated with Peking University Cancer Hospital & Institute and the co-corresponding author. The close collaboration between these institutes is critical for the success of this project.

3. In the author list, who else was CHHP referring to? The CNHPP names should be included in the end of the manuscript or CNHPP should be deleted from author list.

RESPONSE: Corrected as suggested. CNHPP has been deleted from author list and was added in the acknowledgments for funding.

4. In the introduction, page 4, line 7, the 4 subtypes are GC subtypes or DGC subtypes?

RESPONSE: The 4 subtypes are GC subtypes by TCGA. Our study is the first proteomics analysis for DGC subtyping to the best of our knowledge.

5. For spectrum assignment, peptide and protein identifications, the FDR descriptions were not clear, for example, in result session (page 6), 1% FDR was referring to global protein FDR, what is the FDR in peptide and spectrum levels?

RESPONSE: We apologize for not clearly defining FDRs. The FDR of PSMs was 0.08% (peptide spectrum matches) and FDR of peptide was 0.09%. The file used for protein inference and protein FDR calculation was derived from Mascot search results, and we filtered PSM via Percolator and customized parameters. Then proteins were assembled. And the protein FDR was calculated depending on the ratio of NPD (The number of assembled proteins from decoy database searches) and NPT (the number of assembled proteins from target database searches). A flow chart for FDR calculation was provided as follows for better understanding (Figure RL11).

This description was also added in the methods.

Figure RL11. A workflow of FDR calculation.

6. In result session (page 7), the authors described that one of the filters was to pick proteins that were detected in at least 1/6 of the samples (28 cases), what was the rationale for this filter and other filters that were applied to the data analysis.

RESPONSE:

- 1) As shown in Figure RL12, CV (coefficient of variation) of a protein decreased with the increasing number of detected frequencies in the samples. Below the number of 28, CV of a protein is very large. We therefore picked “proteins that were detected in at least 28 samples”, from which CV goes down slowly.

Figure RL12. Each protein’s coefficient of variation (CV) and its detected frequency in 168 samples.

- (2) “3 fold” was used to find proteins that were expressed higher or lower. This threshold was defined using the in-house 293T QC data. We first evaluated the general fold change of proteins from total 240 QC experiments (all were 293T cell profiling data), which were produced by QE-HF3 (120 runs) and Fusion3 (120 runs), respectively. Based on 240 QC experimental data, the density distribution of each protein’s FOT value was plotted using kernel density estimation with a Gaussian kernel and Shafer-Jones bandwidth. Particularly, for proteins with bimodal or tailing distribution of FOT, their representative distribution was extracted from a two-component Gaussian mixture model and had higher mixed probability. The expected FOT value of each protein located in the horizontal coordinate corresponding to the ridge of distribution, then the general fold change of each protein was computed according to the ratio between their the maximum and minimum FOT value in 95% confidence intervals and expected value. The depiction from QC datasets analysis showed general fold change of protein is within 3 folds, which is selected to filter the quantitative data. (Figure RL13-14).

* λ means mixing proportions derived from Gaussian Mixture Model.

Figure RL13. Extracting representative distribution for a protein with bimodal or tailing FOT distribution. Left: Practical FOT distribution of a protein using kernel density estimation with a Gaussian kernel and Shafer-Jones bandwidth. Right: Two fitting distribution derived from left distribution using Gaussian Mixture Model. The fitting distribution with higher mixing proportions is extracted as representative distribution of a protein with bimodal or tailing FOT distribution.

Figure RL14. Distribution plot of protein percentage under increasing fold-change (FC) levels using 240 QC datasets.

(3) When we performed DGC subtyping, we used the filter “with $T/N > 3$ or $T/N < 1/3$ in at least 1/10 of the patients (8 patients)” to pick proteins with greater differential expressions in at least 8 patients. Below the number of 8 patients, lots of proteins with low median absolute deviation (MAD) were involved, so we use the threshold of 8 to remove proteins with low MAD (Figure RL15).

Figure RL15. Density plot for each protein's median absolute deviation (MAD) of $\log_{10}(T-Nratio)$ and the number of patients with $T/N > 3$ or $< 1/3$.

7. In result session (page 7), CVs and FOT of the proteins were analyzed through the samples. These analyses should also performed on the data for the quality control runs in the whole data-collecting period. Is the FOT value consistent throughout the whole data-collecting period? Will different FOT values be used based on the instrument performances or mass spectrometers? This should be assessed by the quality control data.

RESPONSE: We used 293T samples as quality control data, performed on identical mass spectrums in the meantime.

(1) CVs of 293T quality control samples were shown on Figure RL16. When FOT reached more than 10^{-5} , the CV was drastically decreased, suggesting that $FOT > 10^{-5}$ was a good cut-off for accurate quantification.

Figure RL16. Relationship between coefficient of variance (CV) and $\log_{10}(FOT)$. When FOT reached more than 10^{-5} , CV significantly dropped with an average of 9.6%.

(2) For quality control runs, we sorted them by instruments, and the FOT was

consistent (Figure RL17). The FOT was also consistent for 84 tumor and paired normal tissue samples (Figure RL18).

Additionally, since we used T-N ratios of FOT for further analysis, and the tumor and paired normal tissues were run on the same machine at the same time, the small bias due to instrument difference was canceled by taking the T/N ratio (Figure RL19).

Figure RL17. Boxplot for medians of $\log_{10}(\text{FOT})+5$ for 40 293T samples. Different colors denoted different machines.

Figure RL18. Boxplot for $\log_{10}(\text{FOT})+5$ for 84 paired samples with 4 machines. different colors denoted different machines.

Figure RL19. Boxplot for $\log_{10}(T/N)$ for 84 patients with 4 machines. Different colors denoted different machines.

8. In the result session (page 7), 3 folds was selected as cut-off for differentially expressed proteins. Why 3 folds was selected to filter the quantitative data.

RESPONSE: As mentioned in the response to question 6, “3 fold” as the threshold to find proteins differentially expressed was defined using in-house 293T QC data.

Before selecting 3 folds as cut-off for differentially expressed proteins, we evaluated the general fold change of proteins from total 240 QC experiments (all were 293T cell profiling data), which were produced by QE-HF3 (120 runs) and Fusion3 (120 runs), respectively. Based on 240 QC experimental data, the density distribution of each protein’s FOT value was plotted using kernel density estimation with a Gaussian kernel and Shafer-Jones bandwidth. Particularly, for proteins with bimodal or tailing distribution of FOT, their representative distribution was extracted from a two-component Gaussian mixture model and had higher mixed probability. The expected FOT value of each protein located in the horizontal ordinate corresponding to the ridge of distribution, then the general fold change of each protein was computed according to the ratio between their the maximum and minimum FOT value in 95% confidence intervals and expected value. The depiction from QC datasets analysis showed general fold change of protein is within 3 folds, so it is selected to filter the quantitative data. (Figure RL13-14).

* λ means mixing proportions derived from Gaussian Mixture Model.

Figure RL13. Extracting representative distribution for a protein with bimodal or tailing FOT distribution. Left: Practical FOT distribution of a protein using kernel density estimation with a Gaussian kernel and Shafer-Jones bandwidth. Right: Two fitting distribution derived from left distribution using Gaussian Mixture Model. The fitting distribution with higher mixing proportions is extracted as representative distribution of a protein with bimodal or tailing FOT distribution.

Figure RL14. Distribution plot of protein percentage under increasing fold-change (FC) levels using 240 QC datasets.

9. In result session (page 10), the authors stated that “it was remarkable that cancer subtyping and association with clinical outcome could be achieved solely based on the altered cancer proteome.” However, the stability of these subtypes is not confirmed. The subtypes might depend on how the proteomic features were selected.

RESPONSE: In the answer of question 6, we explained why we chose 1) “proteins $>10^{-5}$ in at least 28 samples” to select proteins detected at high frequencies; and 2) proteins “with T/N >3 or T/N $<1/3$ in at least 1/10 of the patients (8 patients)”

to pick proteins with differential expressions, and classify the proteomic subtypes by unsupervised consensus clustering (Figure RL12 and Figure RL15).

To address this reviewer’s concerns, we tested for other thresholds:

- 1) We tested “proteins $>10^{-5}$ in at least 28 samples (1/6)” with “proteins $>10^{-5}$ in at least 17, 21, 42, 56 and 84 samples (1/10~1/2)” to examine the stability of the three subtypes (D5 changed, all other criteria in Figure 1A stayed the same), resulting in total of 2635, 2594, 2538, 2428, 2290 and 2015 proteins, respectively, in each data set. With the thresholds of 1/10, 1/8 and 1/4, no samples were clustered differently; with the thresholds of 1/3 and 1/2, 2 and 7 samples were clustered to different subtypes (Figure RL20).

- 2) We tested “proteins with $T/N > 3$ or $T/N < 1/3$ in at least 1/10 of the patients (8 patients)” with “proteins with $T/N > 3$ or $T/N < 1/3$ in at least 4, 6, 8, 10, 20, 22, 26 or 28 of the patients” to examine the stability of the three subtypes (D6 changed, all other criteria in Figure 1A stayed the same), and the number of proteins used were 3163, 2838, 2538, 2279, 1251, 1110, 856 and 747. With the thresholds from 6 to 20, no samples were clustered differently; With the threshold of 4, 22-26 and 28, the number samples that were clustered to different subtypes was 2, 1 and 4, respectively (Figure RL21).

Together these analyses showed that the subtyping is stable and does not depend strongly on thresholds.

Figure RL20. Clusters assigned by consensus clustering with different number of proteins with different thresholds (1/10, 1/8, 1/6, 1/4, 1/3 and 1/2) of fraction of detected samples.

Figure RL21. Clusters assigned by consensus clustering with different number of proteins with different thresholds (4, 6, 8, 10, 20, 22, 26, 28) of number of samples in which with $T/N > 3$ or $< 1/3$.

10. In result session (page 11), “ Among the 183 mutated genes, only 16 gene products demonstrated altered expressions compared with the wild type (WT), suggesting that mutations do not always strongly impact protein expression.” This statement needs further support or clarification, for example, the lack of protein changes of other mutated genes could due to the small number of samples with the mutations.

RESPONSE: We agree that the small number of mutated samples could lead to no significant protein changes, and we have added this in the discussion section.

Also, tumor heterogeneity should be taken into consideration. If a mutation is present only in a small amount of tumors, the protein level change would not be significant. To answer this question, we increased the threshold of the variant allele (VAT) from 5% to 10, 15 and 20% and then recalculated the respective correlations. For some genes (*RBMX* and *TP53*), increasing of VAT showed expression difference between mutants and wild types, for some genes (*FLI1*, *NF1* and *MED12*), increasing of VAT erased the correlation between mutation and expression, mainly because of smaller mutated sample size. Among all 183 mutated genes we found in this study, very limited number of genes showed positive correlation between mutation status and protein expression, even after increasing of VAT (Table RL1, Figure RL6). Thus, our conclusion is still that the correlation between mutations and protein expression was poor.

Symbol	Variant allele threshold			
	0.05	0.1	0.15	0.2
ARID1A	down	down	down	down
NCOR1	down	down	down	down
ATM	down	down	down	down
PLEKHA6	down	down	down	-
SOX9	down	down	down	-
FLI1	down	-	-	-
BAX	down	down	down	-
RBMX	-	-	down	down
NF1	up	up	-	-
MED12	up	up	-	-
TP53	-	up	up	up

Table RL1. Differentially expressed proteins between mutated samples and wild types with different mutation frequency threshold (“down” denoted mutated samples with lower protein expression, “up” denoted mutated samples with higher protein expression, “-“ denoted no significance).

Figure RL6. Volcano plots to illustrate significantly altered protein expressions for genes with nonsynonymous mutations compared with the wild types (Freq thresholds for (A~D) are 0.05, 0.1, 0.15 and 0.2). Up-regulated and down-regulated genes are indicated in red and blue, respectively. *P* values were calculated using Wilcoxon rank sum test.

11. In method session (page 26), 428 were treated with neoadjuvant chemotherapy or chemo-radiation therapy before operation. The treatment for the samples used in this study should be summarized.

RESPONSE: We apologize for the misleading description. We meant that the 428 patients who were treated with neoadjuvant chemotherapy or chemo-radiation therapy before operation were excluded. The detailed treatment information of all 84 included in this study was listed in **supplementary Data 1**.

12. In method session, were both normal and cancer pieces genomically analyzed?

RESPONSE: Yes, normal and cancer pieces were both genomically analyzed in pairs.

And we added the description of VAT threshold in normal samples in the *Methods*.

13. In method session (page 30), Orbitrap Fusion mass spectrometer and Q Exactive HF mass spectrometer were used to collect MS/MS data, how would different mass spectrometers affect the intensity cut off and spectrum requirement for quantitation?

RESPONSE: As explained in response to question 7, the FOT values were consistent across the whole data collecting period (Figure RL18). Besides the difference was further minimized as the T-N ratios were used for further analysis, for the tumor and normal tissues were dealt with at the same time and in the same machine (Figure RL19).

Figure RL18. Boxplot for $\log_{10}(\text{FOT})+5$ for 84 paired samples with 4 machines. Different colors denoted different machines.

Figure RL19. Boxplot for $\log_{10}(\text{T/N})$ for 84 patients with 4 machines. Different colors denoted different machines.

14. In method session (page 32), what would be the FDR for peptide and spectrum or protein levels? Were FDR applied to the entire assembled data set or

to individual run or sample?

RESPONSE: The FDR described here is the FDR for protein (FDR=0.009), and it was calculated using the entire assembled data set. The FDRs for peptide and PSMs (peptide spectrum matches) is 0.0009 and 0.0008, respectively.

15. The availability of the datasets to the public, and access information of the data should be described in details.

RESPONSE: All raw files and search files have been uploaded onto the iProX (<http://www.iprox.org/>) with identification IPX0001046000. This information was also described in the Data availability in the *Methods*.

Reference

1. Zhang, L., *et al.* Intratumoral T cells, recurrence, and survival in epithelial ovarian cancer. *N Engl J Med* **348**, 203-13 (2003).
2. Galon, J., *et al.* Type, density, and location of immune cells within human colorectal tumors predict clinical outcome. *Science* **313**, 1960-4 (2006).
3. Thompson, E. D., *et al.* Patterns of PD-L1 expression and CD8 T cell infiltration in gastric adenocarcinomas and associated immune stroma. *Gut* **66**, 794-801 (2017).
4. Uhlen, M., *et al.* A pathology atlas of the human cancer transcriptome. *Science* **357**, (2017).
5. Mertins, P., *et al.* Proteogenomics connects somatic mutations to signalling in breast cancer. *Nature* **534**, 55-62 (2016).
6. Zhang, H., *et al.* Integrated Proteogenomic Characterization of Human High-Grade Serous Ovarian Cancer. *Cell* **166**, 755-65 (2016).
7. Wilhelm, M., *et al.* Mass-spectrometry-based draft of the human proteome. *Nature* **509**, 582-7 (2014).
8. Zhang, B., *et al.* Proteogenomic characterization of human colon and rectal cancer. *Nature* **513**, 382-7 (2014).
9. Cancer Genome Atlas Research, N., *et al.* Integrated genomic characterization of oesophageal carcinoma. *Nature* **541**, 169-175 (2017).

REVIEWERS' COMMENTS:

Reviewer #1 (Remarks to the Author):

Although interesting, the revised manuscript unfortunately still remains purely speculative, lacking functional validations in vitro and/or in vivo and validation of clinical association in independent cohort. In addition, majority of the concerns raised by the reviewer was not convincingly addressed. Since proteomic subtypes are only moderately associated with clinical outcomes, their association should be validated independent cohort. Authors speculated that PX3 might be sensitive to immunotherapy. However, many protein overexpressed in PX3 such as IDO1, ARG1, and CD276 are negative regulator of host immune activity. Thus PX3 subtype might be resistant tumors to immunotherapy.

Reviewer #2 (Remarks to the Author):

I am satisfied with the responses to the reviews and the subsequent changes made to the manuscript.

Reviewer #3 (Remarks to the Author):

The revised manuscript has addressed my concerns and comments and is acceptable for publication.

REVIEWERS' COMMENTS:

Reviewer #1 (Remarks to the Author):

Although interesting, the revised manuscript unfortunately still remains purely speculative, lacking functional validations in vitro and/or in vivo and validation of clinical association in independent cohort. In addition, majority of the concerns raised by the reviewer was not convincingly addressed. Since proteomic subtypes are only moderately associated with clinical outcomes, their association should be validated independent cohort. Authors speculated that PX3 might be sensitive to immunotherapy. However, many protein overexpressed in PX3 such as IDO1, ARG1, and CD276 are negative regulator of host immune activity. Thus PX3 subtype might be resistant tumors to immunotherapy.

Response: Functional validations will be our future research. We added “An independent cohort would be required to validate the clinical relevance and subtypes we observed.” in the first paragraph of Discussion part.

Reviewer #2 (Remarks to the Author):

I am satisfied with the responses to the reviews and the subsequent changes made to the manuscript.

Reviewer #3 (Remarks to the Author):

The revised manuscript has addressed my concerns and comments and is acceptable for publication.